# Unsupervised Partner Design Enables Robust Ad-hoc Teamwork

**Constantin Ruhdorfer** [1]  **Matteo Bortoletto** [1]  **Victor Oei** [1]  **Anna Penzkofer** [1]  **Andreas Bulling** [1]

## Abstract

We introduce *Unsupervised Partner Design (UPD)*, a population-free multi-agent reinforcement learning method for robust ad-hoc teamwork. UPD generates training partners on-the-fly and selects them adaptively based on a learnability criterion, removing the need for pre-trained partner populations or manual parameter tuning. We show that this simple mechanism enables effective partner diversity and can be extended to joint partner-environment selection when a procedural level generator is available. Across Level-Based Foraging, Overcooked-AI, and the Overcooked Generalisation Challenge, UPD consistently achieves strong performance compared to both population-based and population-free baselines. In a human-AI user study, agents trained with UPD achieve higher returns and are rated as more adaptive, more human-like, and less frustrating than all evaluated baseline methods.

## 1. Introduction

Robust cooperation with unknown partners, commonly referred to as ad-hoc teamwork (AHT) (Stone et al., 2010), is a core requirement for cooperative artificial intelligence (AI). In AHT settings, agents must coordinate without assumptions about their partners' policies, making learned coordination strategies prone to brittleness when deployment partners differ from those encountered during training.

Thus, training agents for AHT is costly as existing methods typically rely on large populations of diverse partner policies (Strouse et al., 2021; Zhao et al., 2023; Yu et al., 2023; Li et al., 2023b; Wang et al., 2025) or incorporate expert knowledge and hand-crafted models (Albrecht & Ramamoorthy, 2013; Barrett et al., 2014; Albrecht et al., 2016; Barrett et al., 2017). Maintaining and tuning such partner

---

[1]Collaborative Artificial Intelligence, University of Stuttgart, Stuttgart, Germany. Correspondence to: Constantin Ruhdorfer <constantin.ruhdorfer@vis.uni-stuttgart.de>.

*Proceedings of the 43rd International Conference on Machine Learning*, Seoul, South Korea. PMLR 306, 2026. Copyright 2026 by the author(s).

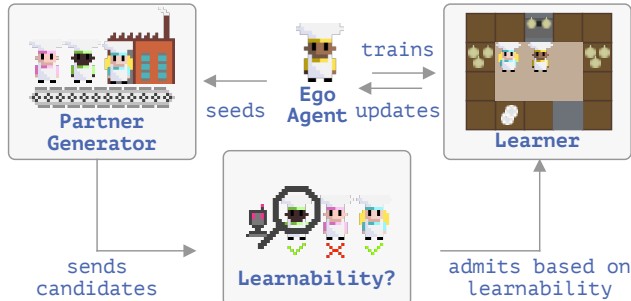

*Figure 1. Unsupervised partner design* is a novel population-free, multi-agent learning framework for ad-hoc teamwork that uses learnability to find training partners for the ego agent to generate an open-ended curriculum.

populations becomes increasingly expensive as tasks and partner diversity scale. Efficient end-to-end training (E3T) (Yan et al., 2023) partially addresses this challenge by generating training partners as stochastic mixtures of an ego and random policy, avoiding explicit partner populations. However, E3T still requires careful tuning of mixture parameters to each task and evaluation setting, limiting its scalability.

In parallel, unsupervised environment design (UED) (Dennis et al., 2020) has shown that adaptive curricula over environment parameters can significantly improve generalisation. Recent work has highlighted that generalising jointly across partners and environments is crucial for robust cooperation, while also showing that existing methods struggle to scale to this setting (Ruhdorfer et al., 2025b).

Our work is guided by two high-level questions. **First**, can collaboration partners be generated cheaply and adaptively – analogous to environment design – and used to train robust ad-hoc teamwork agents without maintaining explicit partner populations? **Second**, does such a partner-design mechanism extend naturally to joint partner-environment curricula in procedurally generated settings, where population-based AHT methods are difficult to scale?

As an answer to both, we introduce **Unsupervised Partner Design (UPD)**, a population-free AHT method that builds an adaptive training distribution over partner behaviours through online generation and selection. UPD removes the need for pre-trained partner populations and hand-tuned mixture coefficients, while retaining strong performance in

zero-shot coordination. When a procedural level generator is available, the same mechanism extends directly to joint partner-environment selection, yielding a simple joint curriculum learning approach. Our core contribution is to show that a simple generation and selection mechanism can result in non-trivial emergent coordination behaviour. Specifically:

1. We show that UED ideas can be extended to the partner space and introduce **Unsupervised Partner Design**, a population-free method for training ad-hoc teamwork agents via adaptive partner generation and selection, eliminating the need for explicit partner populations.

2. We demonstrate that **UPD achieves strong AHT performance** in Level-Based Foraging (Albrecht & Ramamoorthy, 2013) and Overcooked-AI (Carroll et al., 2019), when compared to both population-based and population-free baselines when evaluated with diverse artificial partners and humans in section 5.

3. We show that **the same mechanism extends to joint partner-environment curricula** in procedurally generated settings, achieving robust zero-shot cooperation on the Overcooked Generalisation Challenge (OGC) (Ruhdorfer et al., 2025b) in section 6.

## 2. Related Work

### 2.1. Ad-hoc Teamwork

AHT was explored in a wide range of multi-agent reinforcement learning (RL) environments (Carroll et al., 2019; Bard et al., 2020; Kurach et al., 2020; Ruhdorfer et al., 2025a). Popular AHT methods, such as fictitious co-play (FCP) (Strouse et al., 2021) or maximum entropy population-based training (MEP) (Zhao et al., 2023), rely on pretraining diverse partner populations and optimising best-response policies for these (Yu et al., 2023; Lou et al., 2023; Rahman et al., 2023; Erlebach & Cook, 2024; You et al., 2025). Recent works incorporated open-ended learning objectives to dynamically expand partner diversity (Li et al., 2023b; Wang et al., 2025), but still involved growing partner populations over time or used curricula over pretrained populations (Erlebach & Cook, 2024) or over a partner model learned from offline data (Chaudhary et al., 2025). A notable exception is E3T (Yan et al., 2023), which generates partners on the fly as mixtures of the ego and a random policy. E3T demonstrated strong performance, outperforming prior population-based approaches such as FCP and MEP in human-AI coordination settings. While this approach does not require any partner population and thus significantly reduces the computational overhead, it still requires careful tuning of mixture coefficients between the ego and random policy for each task and evaluation scenario, limiting robustness across settings. In contrast, we propose a lightweight population-free approach that adaptively generates diverse partner behaviour

without fixed parameters or pre-trained populations, and is compatible with existing curriculum learning frameworks.

### 2.2. Unsupervised Environment Design

UED (Dennis et al., 2020) adaptively generates training environments tailored to an agent's capabilities and has proven effective for improving generalisation. Unlike domain randomisation (DR) (Tobin et al., 2017), UED generates environments to target an agent's learning frontier. Existing UED methods mainly focus on single-agent settings and rely on regret-based objectives to guide environment generation (Wang et al., 2019; 2020; Dennis et al., 2020; Jiang et al., 2021b;a; Parker-Holder et al., 2022; Li et al., 2023a; Beukman et al., 2024). Extensions to multi-agent settings are limited: Samvelyan et al. (2023) focused on competitive settings, Ruhdorfer et al. (2025b) proposed a cooperative multi-agent UED benchmark, but no method, while You et al. (2025) trains only with past self-play checkpoints. Recent works reframed UED as a learnability-driven problem, replacing regret-based objectives with scoring functions that directly measure an environment's learning potential (Rutherford et al., 2024; Monette et al., 2025). However, prior work has focused on environment parametrisation only and does not consider partner policies as part of the curriculum space. We extend unsupervised design to partner policies, introducing adaptive partner generation as a population-free curriculum mechanism. When combined with existing UED approaches, this enables joint partner-environment selection in procedurally generated settings for zero-shot cooperation.

## 3. Preliminaries

### 3.1. Reinforcement Learning

Inspired by prior work on multi-agent unsupervised environment design (Samvelyan et al., 2023; Ruhdorfer et al., 2025b), we model our setting as an *under-specified cooperative two-agent stochastic game*. Each environment instance is defined by a Markov game $\mathcal{G}_\theta = \langle S, A, T_\theta, R_\theta, \gamma, \rho_0 \rangle$, indexed by environment parameters $\theta \in \Theta$. The under-specified setting corresponds to a family of levels $\{\mathcal{G}_\theta\}_{\theta \in \Theta}$, where each $\theta$ defines a specific instance of the game (a *level*), for example specifying the locations of walls or objects. At each time step, both agents select actions $a_t^{(1)}, a_t^{(2)} \in A$, the environment transitions according to $s_{t+1} \sim T_\theta(\cdot | s_t, a_t^{(1)}, a_t^{(2)})$, and they receive a shared reward $R_\theta(s_t, a_t^{(1)}, a_t^{(2)})$. $\gamma \in [0, 1]$ is the discount factor. We denote $\tau$ as a trajectory $(s_0, a_0, ..., s_T, a_T)$. As in previous works (Carroll et al., 2019; Yan et al., 2023), we solve the coordination problem using self-play by pairing the ego agent with a fixed partner policy $\pi_p$ and optimise

$$J(\pi_{\text{ego}}, \pi_p) = \mathbb{E}\left[\sum_{t=0}^{\infty} \gamma^t R_\theta(s_t, a_t^{(1)}, a_t^{(2)})\right].$$

In this work, we consider two-agent environments that are both fully specified (LBF, Overcooked-AI) and under-specified, e.g. that feature procedural level generation (the OGC). In the fully specified case, the formalism reduces to a standard two-agent stochastic game. Crucially, this perspective allows us to reason about training over joint distributions of tasks (via $\theta$) and partners (via $\pi_p$).

## 3.2. Ad-hoc Teamwork

Within the under-specified stochastic game formalism above, AHT corresponds to the problem of optimising $\pi_{\text{ego}}$ with respect to an unknown set of partner policies $\Pi_{\text{eval}}$, including humans (Stone et al., 2010). The goal is to find $\pi_{\text{ego}}^* = \arg\max_{\pi_{\text{ego}}} \mathbb{E}_{\pi_p \sim \Pi_{\text{eval}}}[J(\pi_{\text{ego}}, \pi_p)]$. Since $\Pi_{\text{eval}}$ is typically unknown, we cannot solve this problem directly. Many approaches instead use expert domain knowledge or learn a set of diverse partners $\Pi_{\text{train}}$ for training $\pi_{\text{ego}}$ as a best response to $\Pi_{\text{train}}$, via $\pi_{\text{ego}} = \arg\max_{\pi_{\text{ego}}} \mathbb{E}_{\pi_p \sim \Pi_{\text{train}}}[J(\pi_{\text{ego}}, \pi_p)]$. Since $\Pi_{\text{eval}}$ and $\Pi_{\text{train}}$ are typically different, $\pi_{\text{ego}}$ might not be optimal for $\Pi_{\text{eval}}$ because of the distribution shift. However, if $\Pi_{\text{train}}$ sufficiently approximates $\Pi_{\text{eval}}$, $\pi_{\text{ego}}$ can still perform well.

This naturally sets up a two-stage process: training $\Pi_{\text{train}}$ and then training $\pi_{\text{ego}}$ as a best response. Obtaining $\Pi_{\text{train}}$ often comes at considerable costs as querying experts and/or training partners for $\Pi_{\text{train}}$ is expensive. FCP for example trains $N$ policies with different random initialisation using RL to construct $\Pi_{\text{train}}$, while MEP additionally considers interactions between these $N$ policies. If training a policy to completion has cost $C$, then population-based methods incur training costs on the order of $\mathcal{O}(NC)$ or worse due to inter-policy interactions (e.g. MEP).

Opposed to these, E3T is a population-free approach and trains $\pi_{\text{ego}}$ efficiently in self-play, without using a pre-constructed $\Pi_{\text{train}}$, by computing $\pi_p$ as a mixture of $\pi_{\text{ego}}$ and a random policy $\pi_r$:

$$\pi_p = \epsilon \pi_r + (1 - \epsilon)\pi_{\text{ego}}. \tag{1}$$

Here, $\epsilon \in [0, 1]$ is fixed for the training duration. Yan et al. (2023) select $\epsilon$ manually based on the task and evaluation setting, which introduces sensitivity to hyperparameter choices and limits adaptivity across different training stages.

In this work, we consider both the standard AHT problem in which $\Pi_{\text{eval}}$ is unknown but the evaluation cooperation task is known ($\theta_{\text{train}} = \theta_{\text{eval}}$), and the more challenging AHT in unknown levels (Ruhdorfer et al., 2025b; Jha et al., 2025) in which $\pi_{\text{ego}}$ is evaluated in a range of evaluation levels where each level $\theta_{\text{eval}}$ is associated with its own $\Pi_{\text{eval}}$.

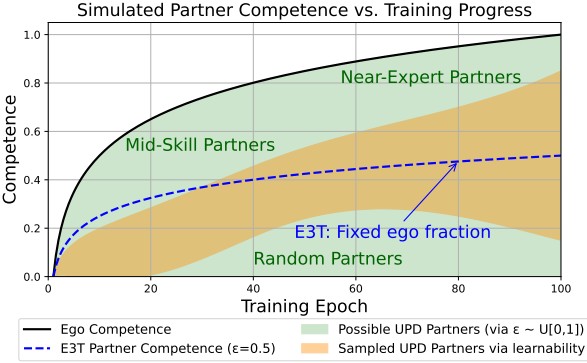

*Figure 2.* **Conceptual illustration:** We illustrate that as ego competence improves over training (black), E3T generates partners using a fixed mixture coefficient (here $\epsilon = 0.5$), resulting in a fixed fraction of ego competence (blue). In contrast, UPD samples $\epsilon \sim \mathcal{U}(0, 1)$ (green) and filters partners using a learnability criterion, leading to a dynamic range of partner competences (orange).

## 3.3. Unsupervised Environment Design

In single-agent RL, UED algorithms use the free parameters of an environment $\theta \in \Theta$ to create a curriculum using a utility function $U$. Many algorithms use regret as the UED objective (Dennis et al., 2020; Samvelyan et al., 2023), where $\theta$ is selected based on the performance difference between the current and an optimal policy: $U(\pi, \theta) = \text{REGRET}_\theta(\pi, \pi_\theta^*)$. However, this assumes access to the optimal policy $\pi_\theta^*$. Recent work has thus moved away from regret as utility. Sampling for learnability (SFL) (Rutherford et al., 2024) scores levels using a learnability function that prioritises instances near the agent's learning frontier. For binary outcomes in which $R(\tau, \theta) \in \{0, 1\}$, learnability is defined as $\ell_{\text{sr}}(\pi, \theta) = p(1 - p)$, where $p = \mathbb{E}_{\tau \sim p(\tau|\pi, \theta)}[R(\tau, \theta)]$ is the success rate on a level. Monette et al. (2025) extended this idea to continuous rewards rewards by weighting return variance around the mean performance. In this work, we apply this idea to *partners* rather than levels, treating them as training instances that can be generated and selected based on learnability.

# 4. Unsupervised Partner Design

*Unsupervised Partner Design* is a population-free learning approach that adaptively generates and selects cooperation partners based on their learnability. The central idea is to apply unsupervised environment design not over environment parameters, but over induced training environments defined by partner policies. Unlike prior work, UPD constructs an adaptive training distribution over partner behaviours through online generation and learnability-guided selection (Figure 2), runs in self-play and only requires a single training stage.

---

**Algorithm 1** Unsupervised Partner Design

---

**Require:** Environment $\mathcal{G}$, ego agent policy $\pi_{\text{ego}}$, partner generator $\mathcal{S}_{\text{p}}$, # scoring rollouts $N$, rollout length $L$, refresh frequency $R$, buffer size $|\mathcal{B}|$, SFL ratio $\rho$
1: Initialize empty buffers $\mathcal{B}, \mathcal{B}_{\text{temp}}; t \leftarrow 0$
2: **while** not converged **do**
3:     **if** $t \bmod R = 0$ **then**
4:         Reset $\mathcal{B}_{\text{temp}} \leftarrow \emptyset$       {Parallelised from here}
5:         **for** each desired partner **do**
6:             Sample $(\pi_p, \theta) \sim \mathcal{S}_p(\pi_{\text{ego}}) \times \Theta$ (Alg. 2)
7:             Get $N$ rollouts of length $L$ in $\mathcal{G}_{\theta}$ for pairing $(\pi_{\text{ego}}, \pi_{\text{p}})$ and collect returns $R_1, \ldots, R_N$
8:             Compute: $\ell_{(\pi_{\text{ego}}, \pi_p, \theta)} = \ell(\pi_{\text{ego}}, \pi_p, \theta)$
9:             Store $(\ell_{(\pi_{\text{ego}}, \pi_p, \theta)}, \pi_{\text{p}}, \theta)$ into buffer $\mathcal{B}_{\text{temp}}$
10:         **end for**
11:         $\mathcal{B} \leftarrow$ top $|\mathcal{B}|$ entries from $\mathcal{B}_{\text{temp}}$ with highest $\ell$
12:     **end if**
13:     Sample a $(\pi_{\text{p}}, \theta)$ batch from $\mathcal{B}$ and $\mathcal{S}_p$ using $\ell$ and $\rho$
14:     Update $\pi_{\text{ego}}$ in $\mathcal{G}_{\pi_p, \theta}$ using PPO
15:     $t \leftarrow t + 1$
16: **end while**

---

## 4.1. Curriculum over Partners

We start from the observation that partners in $\Pi_{\text{train}}$ might not be optimal for learning at a given point in time as, due to its associated cost, $\Pi_{\text{train}}$ is typically fixed to a small number of partners. What if we could instead generate the partners that are useful training instances for $\pi_{\text{ego}}$ throughout training?

For the moment, let us assume access to a diverse partner generator $\pi_p \sim \mathcal{S}_p$ and a fixed level $\theta$. Within the stochastic game formalism introduced above, fixing a partner policy $\pi_p$ induces a training environment for the ego agent by conditioning the co-player in the underlying game:

$$\mathcal{G}_{\pi_p, \theta} := \langle S, A, T_{\pi_p, \theta}, R_{\pi_p, \theta}, \gamma, \rho_{\theta} \rangle, \quad (2)$$

$$T_{\pi_p, \theta}(s'|s, a^{(1)}) = \sum_{a^{(2)}} \pi_p(a^{(2)}|s) T_{\theta}(s'|s, a^{(1)}, a^{(2)}), \quad (3)$$

$$R_{\pi_p, \theta}(s, a^{(1)}) = \sum_{a^{(2)}} \pi_p(a^{(2)}|s) R_{\theta}(s, a^{(1)}, a^{(2)}). \quad (4)$$

Sampling partner policies from $\mathcal{S}_p$ therefore defines a distribution over induced training environments. From the curriculum learning perspective, the goal is to prioritise partners that are likely to induce learning progress in the ego agent. We operationalise this by treating each partner $\pi_p$ as a training instance and scoring it using a learnability function $\ell(\pi_{\text{ego}}, \pi_p)$ estimated from rollout returns. Concretely, we sample multiple partners using $\mathcal{S}_p$ and score them in $\mathcal{G}_{\pi_p, \theta}$:

$$\ell_{\text{var}}(\pi_{\text{ego}}, \pi_p, \theta) = \text{Var}_{\tau \sim \mathcal{G}_{\pi_p, \theta}}[R(\tau)]. \quad (5)$$

Low return variance indicates either consistent failure or success, whereas high variance corresponds to partners of

---

**Algorithm 2** Partner Policy Generator $\mathcal{S}_{\text{p}}$

---

**Require:** Ego policy $\pi_{\text{ego}}$, Bias prob. $p_{\text{bias}} = 0.5$
1: Sample mixing parameter $\epsilon \sim \mathcal{U}(0, 1)$
2: With probability $p_{\text{bias}}$:
3:     Sample persistent bias mask $m \sim \text{Dirichlet}(\alpha \cdot \mathbf{1}_A)$
4: Otherwise:
5:     Set $m = \mathbf{1}_A / A$
6: Obtain biased random policy $\pi_{r,m}$ with bias mask $m$
7: Return partner policy $\pi_p = \epsilon \pi_{r,m} + (1 - \epsilon) \pi_{\text{ego}}$

---

intermediate difficulty where cooperation sometimes succeeds. Intermediate difficulty samples are known to promote learning (Florensa et al., 2018; Tzannetos et al., 2023).

Recent work provides a formal connection between learnability and expected policy improvement. In particular, Foster et al. (2026) show that for a broad class of advantage-based policy gradient methods (including PPO) the expected improvement of the policy is proportional to the variance of the scalar learning signal used to form the advantage. In our setting, fixing a partner policy $\pi_p$ and level $\theta$ induces a single-agent game $\mathcal{G}_{\pi_p, \theta}$ for the ego agent. Applying the result of Foster et al. to this induced game implies that the expected one-step improvement of $\pi_{\text{ego}}$ when training in $\mathcal{G}_{\pi_p, \theta}$ increases with $\text{Var}_{\tau \sim \mathcal{G}_{\pi_p, \theta}}[R(\tau)]$. Therefore, return variance provides a principled signal for prioritising partners that are expected to induce policy improvement in the ego agent. In this sense, UPD extends the curriculum mechanism of SFL from levels to partner policies.

## 4.2. Curriculum over Partners and Levels

Generalising to both novel partners and levels is critical for robust cooperation in under-specified environments (Ruhdorfer et al., 2025b). Notably, UPD readily extends to joint curricula over both partners and levels in an underspecified game. In this case, $\ell$ is simply calculated by sampling a tuple $(\pi_p, \theta) \sim \mathcal{S}_p \times \Theta$ and obtaining $\tau \sim p(\tau|\pi_{\text{ego}}, \pi_p, \theta)$. One problem in this formalism is that different levels $\theta$ might induce different reward ranges. To address this, we use a coefficient-of-variation squared ($\text{CV}^2$) score to correct $\ell$:

$$\ell_{CV^2}(\pi_{\text{ego}}, \pi_p, \theta) = \frac{\text{Var}_{\tau \sim (\pi_{\text{ego}}, \pi_p, \theta)}[R(\tau, \theta)]}{\left(\mathbb{E}_{\tau \sim (\pi_{\text{ego}}, \pi_p, \theta)}[R(\tau, \theta)] + \delta\right)^2}, \quad (6)$$

where $\delta$ is a small constant. We refer to this extension as Joint UPD (JUPD), and present a method sketch in Alg. 1.

## 4.3. Online Partner Generation

To instantiate the curriculum over partners, we require a partner policy generator $\mathcal{S}_p$ from which candidate partners can be sampled (see Alg. 2). In this work, we extend the E3T partner generation strategy by introducing stochasticity

along both competence and behavioural dimensions. However, UPD can be used with *any* form of partner generator.

E3T relies on a fixed mixing coefficient $\epsilon$, which can be suboptimal across different stages of training. Early in learning, more competent partners can improve task learning, whereas later, less predictable partners pose challenges. Rather than fixing $\epsilon$, UPD samples $\epsilon \sim \mathcal{U}(0, 1)$ throughout training, generating partners that span a broad range of competencies.

Beyond competence variation, cooperative partners often exhibit systematic behavioural tendencies. For instance, human players in Overcooked display persistent action preferences, such as favouring particular movement directions or overusing the `stay` action (Carroll et al., 2019; Yu et al., 2023). To capture such low-level biases, UPD introduces a bias masking mechanism. When generating a partner, we sample a persistent bias mask $m \sim \mathrm{Dir}(\alpha \cdot \mathbf{1}_A)$, which defines a biased random policy over the discrete action space. The Dirichlet distribution provides a simple way to control the strength of such biases via a single parameter $\alpha$.

### 4.4. UPD and Convention Selection

In coordination games, self-play converges to one of multiple equivalent equilibria which yields high self-play performance but poor partner generalisation (Carroll et al., 2019; Hu et al., 2020). Consider the $2 \times 2$ matrix game:

$$\begin{array}{c|cc} & \texttt{up} & \texttt{down} \\ \hline \texttt{up} & 1 & 0 \\ \texttt{down} & 0 & 1 \end{array} . \quad (7)$$

Self-play selects only one equilibrium, e.g. $\pi_{\mathrm{SP}}^{\mathrm{up}}$, yielding

$$\mathbb{E}_{\tau \sim (\pi_{\mathrm{SP}}^{\mathrm{up}}, \pi_{\mathrm{SP}}^{\mathrm{up}})}[R(\tau)] = 1, \ \mathrm{Var}_{\tau \sim (\pi_{\mathrm{SP}}^{\mathrm{up}}, \pi_{\mathrm{SP}}^{\mathrm{up}})}[R(\tau)] = 0. \quad (8)$$

In contrast, suppose UPD's partner generator $\mathcal{S}_p$ samples at least one partner $\pi_p$ with $0 < p(\texttt{up}) < 1$. Then

$$\mathbb{E}_{\tau \sim (\pi_{\mathrm{SP}}^{\mathrm{up}}, \pi_p)}[R(\tau)] < 1, \ \mathrm{Var}_{\tau \sim (\pi_{\mathrm{SP}}^{\mathrm{up}}, \pi_p)}[R(\tau)] > 0, \quad (9)$$

with $\ell(\pi_{\mathrm{SP}}^{\mathrm{up}}, \pi_p) > \ell(\pi_{\mathrm{SP}}^{\mathrm{up}}, \pi_{\mathrm{SP}}^{\mathrm{up}})$ which biases UPD toward training with partners that do not fully share this convention. Furthermore, in this game, for a policy $\pi_{\mathrm{SP}}^{\mathrm{up}}$ and a partner $\pi_p$ with $p = \pi_p(\texttt{up})$, $\ell_{\mathrm{var}}$ is maximized at $p = 0.5$. Hence, if the partner generator $\mathcal{S}_p$ has support near $p = 0.5$[1], UPD will select a partner that is maximally ambiguous and thus can introduce gradient pressure away from the current equilibrium, discouraging overfitting to a convention while also maximising expected policy improvement (subsection 4.1).

---

[1] For the partner generator used in this work (Alg. 2) and fixed $\pi_{\mathrm{SP}}^{\mathrm{up}}$, $\ell$ is maximized by $\epsilon = 1.0$ and $p_{\mathrm{bias}} = \texttt{False}$, which yields a uniform random partner with $p(\texttt{up}) = 0.5$. This setting would be discovered in expectation by sufficiently sampling from $\mathcal{S}_p$. E3T, in contrast, would require $\epsilon$ being set optimally manually which is possible for this example but not generally. Moreover, since the optimal $\epsilon$ depends on the task $\theta$ (as in this example), E3T's fixed strategy fails to provide optimal partners across levels.

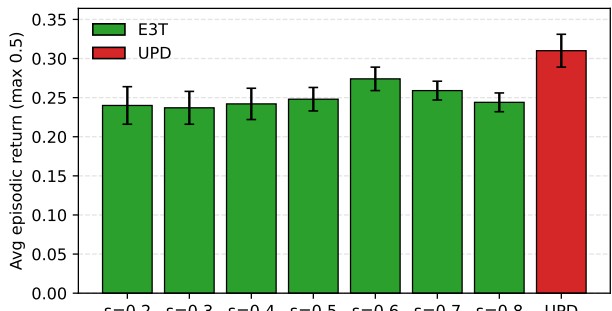

*Figure 3.* Average returns in cooperative LBF with ten evaluation partners. Bars show mean $\pm$ standard deviation. UPD achieves higher average returns than E3T across all tested $\epsilon$.

## 5. Experiments in Fixed Environments

We split our experiments into two sections. **In this section, we evaluate UPD in environments without procedural generation** where $\theta$ is fixed, using Level-Based Foraging and Overcooked-AI. Our goal here is to assess UPD's effectiveness as an AHT method. In section 6, we turn to **procedurally generated environments using OGC**, evaluating whether the same mechanism extends to joint generalisation across partners and levels within this setting. In total, our paper evaluates 282 trained policies.

### 5.1. Comparing UPD and E3T in LBF

We first analyse the behaviour of the current population-free baseline, E3T, by comparing it to UPD on LBF (Albrecht & Ramamoorthy, 2013) – a popular environment in AHT research (Rahman et al., 2021; Papoudakis et al., 2021; Mirsky et al., 2022; Rahman et al., 2023; Wang et al., 2025). We base our experiments on the version used in (Bonnet et al., 2024; Wang et al., 2025), where two agents need to work together to collect three foods in a $7 \times 7$ grid, each food requiring both agents to be loaded. The game terminates after 100 steps, when agents collide or when all food has been collected, yielding a maximum return of $0.5$. Since E3T has not been used on LBF yet, $\epsilon$ is determined empirically by sweeping $\epsilon \in \{0.2, 0.3, \ldots, 0.8\}$.

To construct a diverse evaluation population $\Pi_{\mathrm{eval}}$, similar to Wang et al. (2025), we combine hardcoded agents, planning models, and agents trained using BRDiv (Rahman et al., 2023). We use three BRDiv agents, one random, and six planning agents (collecting food in (reverse) column-major, (reverse) lexicographic, (reverse) nearest-to-farthest) for a total of ten evaluation partners. A BRDiv population optimises self-play while minimising cross-play returns and thus adopts competent but incompatible strategies. Note that not all of these partners act optimally; optimal cooperation with those might still result in below-maximum returns.

*Table 1.* Average returns (mean $\pm$ std.) with evaluation populations in Overcooked-AI. We average over both starting positions. The best results are in **bold**, second-best are underlined. UPD outperforms the considered baselines in aggregate.

| Method | CRoom | AA | CR | CC | FC | Average | % Gain rel. to E3T |
|---|---|---|---|---|---|---|---|
| SP | 68.9$\pm$ 5.6 | 64.4$\pm$ 2.3 | 25.3$\pm$ 4.5 | 14.5$\pm$10.2 | 33.9$\pm$8.3 | 41.4$\pm$ 4.7 | -62.2% |
| FCP | 109.9$\pm$ 5.8 | 117.4$\pm$ 7.0 | 64.7$\pm$ 3.0 | 30.8$\pm$ 9.7 | 27.2$\pm$4.8 | 70.0$\pm$ 2.2 | -11.8% |
| MEP | 109.5$\pm$ 5.3 | 142.8$\pm$36.1 | 64.0$\pm$ 2.2 | 13.7$\pm$ 5.3 | 46.2$\pm$6.1 | 75.3$\pm$ 7.6 | -4.5% |
| E3T | 111.3$\pm$7.8 | 127.9$\pm$22.1 | 58.4$\pm$ 4.5 | 55.0$\pm$ 2.4 | 41.3$\pm$8.3 | 78.8$\pm$12.5 | - |
| UPD w/o bias | **111.5$\pm$2.3** | 159.9$\pm$20.7 | 66.2$\pm$ 3.8 | 57.1$\pm$ 2.8 | 44.4$\pm$5.5 | 87.9$\pm$ 3.6 | +10.9% |
| UPD w/o $\ell$ | 110.8$\pm$ 3.1 | 164.0$\pm$18.7 | 68.9$\pm$ 3.4 | **64.5$\pm$3.9** | 45.7$\pm$3.8 | 90.8$\pm$ 6.0 | +14.1% |
| **UPD (Ours)** | 107.5$\pm$ 2.8 | **181.4$\pm$6.4** | **69.2$\pm$5.6** | **64.5$\pm$1.3** | **48.7$\pm$2.8** | **94.4$\pm$2.3** | +18.0% |

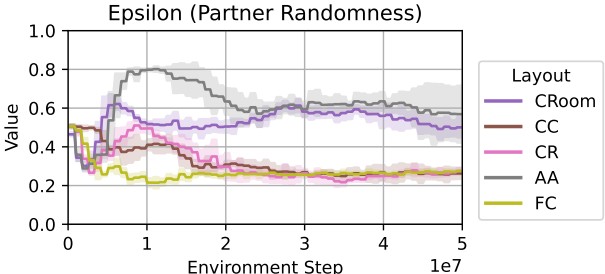

*Figure 4.* We compare how $\ell_{\text{var}}$ selects different average $\epsilon$ values across layouts during training. For layouts that are known to feature narrower coordination challenges (CC, CR & FC), UPD favours smaller, while for CRoom and AA, UPD favours higher $\epsilon$.

We train 6 seeds for $10^7$ timesteps, use learning rate 1e$-$3, use 512 environments (400 steps/env), UPD uses $|\mathcal{B}| = 64$, $R = 4$, $N = 5$, $p_{\text{bias}} = 0.5$, and $\alpha = 1.0$. Figure 3 shows that no tested $\epsilon$ achieves the highest cooperation performance, whereas UPD consistently yields higher returns with $\Pi_{\text{eval}}$. This illustrates a practical limitation of E3T in this setting: fixing $\epsilon$ restricts the range of partner behaviours encountered during training (cf. Figure 2). **UPD addresses this limitation**, and its adaptive range of partner behaviours provides the strongest returns. In practice, sweeping $\epsilon$ may be infeasible in larger settings, while UPD avoids this tuning altogether with only marginal additional runtime ($< 10\%$ in our experiments; we characterise UPD's efficiency in Appendix F).

### 5.2. Overcooked-AI with Artificial Partners

#### 5.2.1. EXPERIMENTAL SETUP

To evaluate UPD under richer coordination dynamics, we next consider Overcooked-AI (Carroll et al., 2019) with the five standard layouts: Cramped Room (CRoom), Asymmetric Advantages (AA), Coordination Ring (CR), Counter Circuit (CC), and Forced Coordination (FC). We follow the same protocol for generating $\Pi_{\text{eval}}$ as for LBF. Concretely, $\Pi_{\text{eval}}$ contains: (1) 3-4 BRDiv agents, (2) probabilistic planning agents (Wang et al., 2025), (3) hardcoded agents performing task subsets (e.g., onion-only

workers) (Yu et al., 2023), and (4) low-competence random and stay agents. CRoom uses nine such partners, AA, CR, and CC use eight, and FC uses four due to its restrictive dynamics that limit viable hardcoded strategies.

We compared UPD against four baselines that, together, cover the most widely and successfully used AHT algorithms in Overcooked-AI: (1) a self-play baseline trained using IPPO (de Witt et al., 2020), (2) E3T using their reported hyperparameters: $\epsilon = 0.5$ for CRoom, AA, CR, and CC, and $\epsilon = 0.0$ for FC, (3) MEP agents using a population size of 48, and (4) FCP agents trained with a population of size 48. To ensure competitive baselines, we use larger population sizes for MEP than in the original works, which improves performance (Yu et al., 2023) and tune MEP populations per layout if required. In contrast, we found that UPD works with one set of hyperparameters on all layouts.

We used the same recurrent policy network for all methods. For E3T and UPD, we added the partner model proposed by E3T. Training used 512 envs (400 steps/env), $5 \times 10^7$ total time steps, with reward shaping for the first $3 \times 10^7$. For UPD, we used a partner buffer size of $|\mathcal{B}| = 512$, $N = 10$ evaluation rollouts per partner, Dirichlet parameter $\alpha = 1.0$, and refreshed the buffer every fourth training loop ($R = 4$). We train and report results from 6 seeds. All methods converge stably, see Figure 10 - Figure 14.

#### 5.2.2. RESULTS

Table 1 shows that UPD achieves the highest average return when paired with diverse, unseen partners, outperforming the considered population-based and population-free baselines in this evaluation. Other methods achieve lower average returns but outperform UPD on CRoom. Our takeaway is not that UPD dominates every layout, but that a single population-free configuration remains competitive or better than strong population-based methods while avoiding explicit partner training.

We also compared with two ablations in Table 1 (bottom): *UPD w/o bias* uses a uniformly random policy $\pi_r$ for partner mixing, while *UPD w/o $\ell$* removes learnability scoring and instead samples random partners per rollout. Each variant

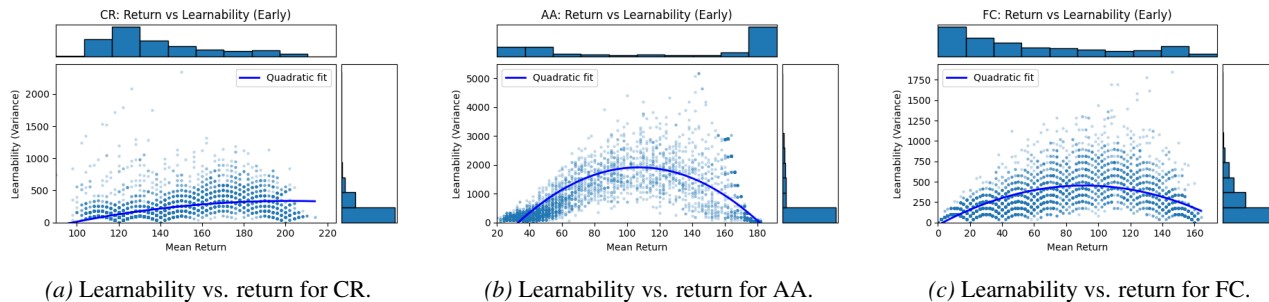

*(a)* Learnability vs. return for CR.   *(b)* Learnability vs. return for AA.   *(c)* Learnability vs. return for FC.

*Figure 5.* We plot learnability vs. return in three representative layouts: CR, AA, and FC. Each dot represents a single potential partner. The bar plots on the axis count partners in their respective return/learnability bands. UPD identifies partners of intermediate difficulty.

still produces competitive AHT agents, but their combination in full UPD yields the best results. Interestingly, *UPD w/o ℓ* performs quite well, likely because the online partner generator alone induces a natural curriculum by sampling a broad spectrum of partners. However, UPD outperforms *UPD w/o ℓ* on average and particularly in AA, where learnability has the strongest impact. A large share of the AHT improvement thus appears to come from large-scale partner generation and biasing, with learnability providing an additional gain. Together, we find that UPD outperforms all ablations on average, underscoring the benefits of dynamic, learnability-driven curricula.

### 5.2.3. ANALYSIS OF PARTNER SELECTION DYNAMICS

We now analyse how UPD selects partners during learning. Figure 4 shows that UPD favours different average $\epsilon$ values across layouts and time steps. In the figure, we average over the $\epsilon$ values in the buffer. In layouts with narrower cooperation conventions (CR, CC, FC), UPD gravitates towards lower $\epsilon$; in more flexible layouts (CRoom, AA), it prefers more stochastic partners. **This indicates that learnability favours different ranges of partner competence across layouts and time** and that no single $\epsilon$ would be optimal.

**Training dynamics under UPD**   To further investigate the role learnability plays in the training dynamics, we plot learnability vs. return for all generated partners in Figure 5, similar to prior work (Rutherford et al., 2024; Monette et al., 2025). We use three representative layouts (CR, AA, and FC), and evaluate the trained ego agent after 40% of training by rolling out with 8,192 randomly generated partners. We select these three layouts as they are representative: both CRoom and CC behave very similarly to the CR case. Across these, we observe three major phenomena: (1) learnability is zero both for partners with whom the ego agent performs well as well as comparatively poorly, (2) learnability is highest for partners with intermediate difficulty, and (3) learnability is high for very few of the generated partners: most partners receive a learnability score of zero or close to

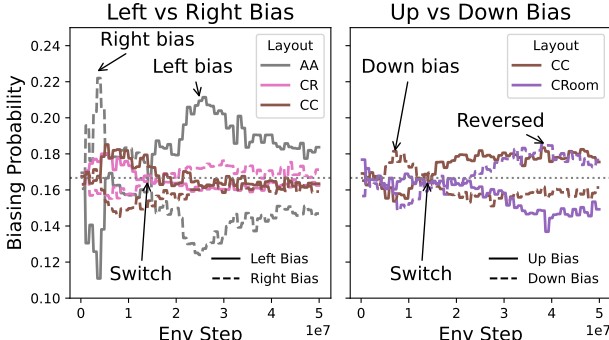

*Figure 6.* Action biases added in partner generation over training. We find that **UPD induces emergent convention breaking**, with partners initially biased toward one action before switching.

zero. This is consistent with the motivation in Sec. 4.1: most partners are not equally beneficial, and **UPD consistently identifies rare partners of intermediate difficulty**.

**Emergent Convention Breaking**   Figure 6 shows systematic shifts in average directional action biases among generated partners during training: early on, partners often exhibit a consistent directional preference (e.g., favouring `right`), which later switches to the opposite bias. This behaviour is consistent across layouts and seeds. These bias switches are consistent with learnability favouring partners that violate the ego agent's current coordination conventions, as such partners tend to induce higher return variability (see subsection 4.4). Unlike prior zero-shot coordination approaches that rely on hand-engineered convention-breaking mechanisms (Hu et al., 2020), UPD exhibits analogous dynamics without any explicit convention-breaking components.

### 5.3. Human-AI Experiments in Overcooked-AI

As a final experiment on Overcooked-AI, we evaluated how UPD performs with humans in a double-blind user study. We evaluated four representative agents – SP, MEP, E3T, and UPD – on three layouts with challenging coordination

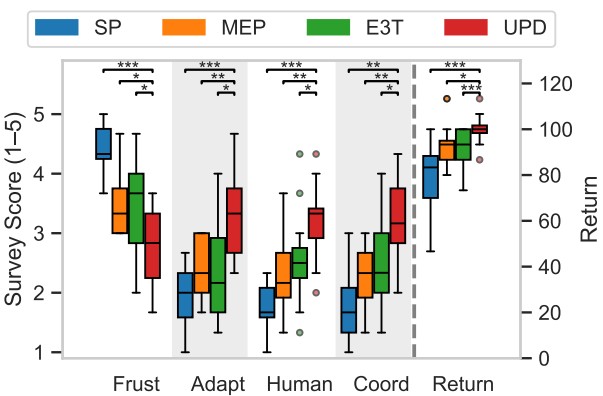

*Figure 7.* Human evaluation of partners: Frust = 'frustrating?' (↓), Adapt = 'adapted well?' (↑), Human = 'human-like?' (↑), Coord = 'coordinated well?' (↑). We performed one-sided Wilcoxon signed-rank tests on individual survey questions and one-sided paired t-tests on return, comparing UPD to each baseline. Significance levels are Holm-Bonferroni-corrected to account for three comparisons ($* = p < 0.05$, $** = p < 0.01$, $*** = p < 0.001$).

dynamics: AA, CR, and CC. Twelve participants (ages 22-31, five female) were recruited, each playing all agent-layout combinations in randomised order. We collected a total of 144 games, with 36 games played per method (200 steps per game, following (Jha et al., 2025)). A tutorial session preceded the trials, and compensation was provided per institutional ethics guidelines. The institutional ethics review board approved the study.

Figure 7 shows that **UPD achieves higher average returns than the considered baselines in human-AI interactions** (right), and is also preferred across most subjective survey items (left): participants found **UPD significantly more adaptive, more human-like, a better collaborator, and less frustrating to work with**. To evaluate overall subjective preference, we aggregated individual responses across the survey items. The ratings showed high internal consistency (Cronbach's $\alpha = 0.916$), justifying the use of a composite score. UPD received higher aggregate preference scores than other methods and their differences were statistically significant (Wilcoxon signed-rank tests, Holm-Bonferroni-corrected, $p < 0.05$). Given the sample size, these results demonstrate that UPD is effective at collaborating with humans, rather than establishing it as the best-performing method overall.

## 6. Experiments with Procedural Generation

A central motivation for UPD is its compatibility with UED, enabling joint curricula over both partners and levels. To test this, we evaluated UPD within the OGC (Ruhdorfer et al., 2025b) – a particularly difficult environment, where levels are randomly generated and not guaranteed to be

*Table 2.* Results (mean ± std.) for joint unsupervised environment and partner generalisation on the 5×5 Overcooked Generalisation Challenge. Best results are **bold**, second-best are underlined.

| Method | CRoom | CR | FC | Avg. |
|---|---|---|---|---|
| DR-DR | 86.4±7.2 | 53.6±8.3 | 9.7±2.4 | 49.9±3.2 |
| CEC | 41.4±9.2 | 20.3±7.0 | 12.7±3.5 | 23.9±9.7 |
| SFLE3T | 81.7±7.8 | 41.1±10.1 | 7.4±2.5 | 44.0±3.0 |
| **JUPD** | **97.0±7.4** | **60.0±6.1** | **17.1±4.2** | **58.9±4.4** |

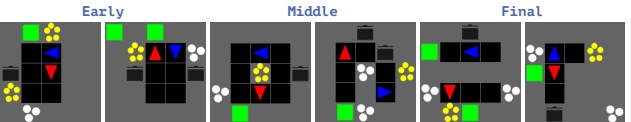

*Figure 8.* Examples of levels sampled into the JUPD training buffer at different stages of training.

solvable, placing a heavy burden on the UED mechanism. In OGC, $\Theta$ varies the locations of walls, objects, and agents. To ensure tractability, we used the 5×5 version of the OGC and trained all methods for $10^9$ steps using 1,024 parallel environments. This is required for convergence in OGC due to the high variance induced by procedural level generation. Evaluation was performed using held-out level-partner combinations and held-out layouts with adjusted evaluation populations matched to the Overcooked-AI experiments.

Classic AHT methods are not designed to scale to procedurally generated environments (Ruhdorfer et al., 2025b; Jha et al., 2025). Because of this, we compared JUPD against several baselines that combine level and partner generation: (1) **DR-DR** randomly samples partners and levels, (2) a **Cross-Environment-Cooperation (CEC)** adaption that selects levels at random and plays in self-play. CEC performs well when the training level generator closely matches the evaluation layouts (Jha et al., 2025). (3) **SFLE3T** combines the best of the literature and selects levels using SFL and creates E3T partners with $\epsilon = 0.5$.

As shown in Table 2, JUPD achieves the highest average return across all evaluated layouts. Compared to DR-DR and CEC, which do not adapt partner difficulty, and SFLE3T, which relies on static partner generation, JUPD jointly selects suitable partners and levels during training. Sampled levels over time are visualised in Figure 8.

## 7. Discussion and Limitations

The experiments in this paper were motivated by two questions: whether collaboration partners can be designed cheaply and adaptively, analogous to level design, and whether such a mechanism extends naturally to joint partner-environment curricula.

Because our goal is to design an AHT method for the joint

curriculum setting, we focus on a partner generator that cleanly extends to procedurally generated environments via learnability and by relying only on the ego policy. In this setting, pre-trained partner populations may not transfer to unseen environment instances, making online partner generation a natural choice.

Since UPD w/o $\ell$ already performs strongly, one practical implication of our work is that extending E3T with randomised mixture coefficients and large-scale parallel partner generation may provide a stronger population-free baseline than standard E3T.

More generally, UPD can be understood as a framework that operates over a partner space. In our experiments, this is instantiated using SFL and the space is defined by a stochastic generator, but in principle other UED methods could be used and the partner space could also be defined by partner populations or latent partner spaces (Liang et al., 2024) when these are available. Exploring richer partner spaces within our framework is an interesting direction for future work. While UPD avoids explicit population pretraining, it shifts computation toward large-scale online partner evaluation. In practice this tradeoff is favourable in highly vectorised simulators such as JAX environments, but may be less advantageous in settings where environment interaction is expensive.

## 8. Conclusion

We introduced Unsupervised Partner Design (UPD), a population-free method for training ad-hoc teamwork agents via learnability-based partner selection. Across multiple benchmarks and a human study, UPD achieves strong AHT performance without pre-trained partner populations or manual parameter tuning. These results suggest that adaptive partner curricula provide a practical and scalable approach to improving generalisation in cooperative multi-agent systems.

## Impact Statement

This work seeks to improve methods for training agents that can cooperate with previously unseen partners, especially humans. While our contributions are evaluated in controlled research environments, improved human–AI coordination may eventually be applied in real-world interactive systems. Here, possible deployment issues that could arise from human-AI coordination failures or misalignment require careful consideration. These concerns are not unique to our approach and are shared broadly across research on interactive and collaborative AI systems. We do not foresee specific new ethical risks introduced by this work beyond those commonly associated with human–AI interaction.

## Acknowledgements

We thank the anonymous reviewers for their feedback, especially for suggesting to additionally compare to ROTATE and for suggesting to extend our discussion on UPD and convention selection. The authors also thank the International Max Planck Research School for Intelligent Systems (IMPRS- IS) for supporting C. Ruhdorfer and V. Oei.

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

## A. Infrastructure & Tools

We ran our experiments on a server system equipped with NVIDIA H100-NVL GPUs with 94 GB of memory and AMD EPYC 9454 CPUs. All training runs were executed on a single GPU. We trained our models using Jax (Bradbury et al., 2018), Flax (Heek et al., 2023) and Optax (DeepMind et al., 2020). We performed our analyses using NumPy (Harris et al., 2020), Pandas (Team, 2020), SciPy (Virtanen et al., 2020) and Matplotlib (Hunter, 2007). Our single-file IPPO implementation was based on the one provided by JaxMARL (Rutherford et al., 2023). Our basis for the FCP, E3T, and by extension MEP and UPD implementations was the published code of Jha et al. (2025). ROTATE and evaluation partners were generated using the code of Wang et al. (2025). Our curriculum learning algorithm is SFL and we base our code on the open-source release of Rutherford et al. (2024).

Our experiments only required a fraction of the computing power that the system above offers. At a minimum, our experiments are reproducible using GPUs with 16 GB of memory, possibly even less. Individual Overcooked-AI training runs typically run on the order of minutes to hours depending on the method: Population-based baselines and ROTATE are more expensive due to pretraining/open-ended iterations. OGC experiments usually take a couple of hours but finish in well under a day.

Our human-AI experiments were conducted using NiceWebRL (Carvalho et al., 2025).

## B. Reproducibility Statement

To reproduce our work, we provide all key hyperparameters in this document. Our project page is available at `https://git.hcics.simtech.uni-stuttgart.de/public-projects/UPD`.

**Transparency Note** An earlier version of this paper contained implementation and configuration issues affecting baseline experiments. After correcting and rerunning the affected experiments, baseline performance improved, but the main conclusions of the paper remained unchanged.

## C. UPD and Convention Selection Extended

Here, we extend the discussion in Section 4.4. Section 4.4 illustrates conceptually how the learnability formulation and convention selection interact. Assume a matrix game such as the following:

$$
\begin{array}{c|cc}
 & \text{up} & \text{down} \\
\hline
\text{up} & 1 & 0 \\
\text{down} & 0 & 1
\end{array}.
\tag{10}
$$

Here, naive self-play would converge on one equilibrium because it maximises the self-play objective $J_{\text{SP}}(\pi)$ which admits two equally good but mutually exclusive solutions: $\pi_{\text{SP}}^{\text{up}}$ and $\pi_{\text{SP}}^{\text{down}}$ where both agents always play up or down respectively. They are mutually incompatible since

$$
\mathbb{E}_{\tau \sim (\pi_{\text{SP}}^{\text{up}}, \pi_{\text{SP}}^{\text{up}})}[R(\tau)] = \mathbb{E}_{\tau \sim (\pi_{\text{SP}}^{\text{down}}, \pi_{\text{SP}}^{\text{down}})}[R(\tau)] = 1
\tag{11}
$$

but

$$
\mathbb{E}_{\tau \sim (\pi_{\text{SP}}^{\text{up}}, \pi_{\text{SP}}^{\text{down}})}[R(\tau)] = 0.
\tag{12}
$$

The goal in zero-shot coordination and ad-hoc teamwork is to learn policies that generalise across different partners and to specifically avoid overfitting to a single equilibrium. UPD does this via two interacting mechanisms: by creating partners based on the current $\pi_{\text{ego}}$ and by selecting partners using a learnability score $\ell$ that is based on the variance of outcomes. Assume, for the sake of argument, that we now perform the UPD update operator on $\pi_{\text{SP}}^{\text{up}}$ (i.e. $\pi_{\text{ego}} = \pi_{\text{SP}}^{\text{up}}$) and that UPD creates three new partners via $\mathcal{S}_p$:

$$
\pi_p^{\text{up}} = \{a | p(a) = 1 \text{ if } a = \text{up else } 0\}
\tag{13}
$$

$$
\pi_p^{\text{down}} = \{a | p(a) = 1 \text{ if } a = \text{down else } 0\}
\tag{14}
$$

$$
\pi_p^{\text{mix}} = \{a | p(a) = 0.5 \text{ if } a = \text{up else } 0.5\}
\tag{15}
$$

It is clear that $\text{Var}_{\tau \sim (\pi_{\text{SP}}^{\text{up}}, \pi_p^{\text{up}})}[R(\tau)] = \text{Var}_{\tau \sim (\pi_{\text{SP}}^{\text{up}}, \pi_p^{\text{down}})}[R(\tau)] = 0$. This aligns with the intuition presented in Section 4.1: $\pi_p^{\text{up}}$ as a partner is too easy, since the ego is already performing optimally with them. Analogously, since

*Table 3.* We additionally compare with ROTATE (Wang et al., 2025) and a fine-tuned version of UPD with curriculum hyperparameters picked per Overcooked layout (see Table 5). Average returns (mean $\pm$ std.) with evaluation populations in Overcooked-AI. We average over both starting positions. The best results are in **bold**, second-best are underlined.

| Method | CRoom | AA | CR | CC | FC | Average | % Gain rel. to E3T |
|---|---|---|---|---|---|---|---|
| ROTATE | **119.9$\pm$7.9** | 147.7$\pm$22.6 | **80.5$\pm$7.2** | 58.1$\pm$ 0.1 | **52.1$\pm$5.1** | 91.7$\pm$ 6.8 | +15.1% |
| UPD (Ours) | 107.5$\pm$ 2.8 | 181.4$\pm$6.4 | 69.2$\pm$ 5.6 | 64.5$\pm$1.3 | 48.7$\pm$2.8 | 94.4$\pm$2.3 | +18.0% |
| UPD (fine-tuned) | 111.6$\pm$7.7 | **201.8$\pm$3.0** | 79.6$\pm$5.1 | **66.7$\pm$1.1** | 51.3$\pm$1.3 | **102.3$\pm$0.6** | +25.9% |

$\mathbb{E}_{\tau \sim (\pi_{\mathrm{SP}}^{\mathrm{up}}, \pi_p^{\mathrm{down}})}[R(\tau)] = 0$ across all possible testing games, $\pi_p^{\mathrm{down}}$ is currently incompatible. Here, UPD would pick $\pi_p^{\mathrm{mix}}$ since $\mathrm{Var}_{\tau \sim (\pi_{\mathrm{SP}}^{\mathrm{up}}, \pi_p^{\mathrm{mix}})}[R(\tau)] = 0.25 > \mathrm{Var}_{\tau \sim (\pi_{\mathrm{SP}}^{\mathrm{up}}, \pi_p^{\mathrm{up}})}[R(\tau)] = \mathrm{Var}_{\tau \sim (\pi_{\mathrm{SP}}^{\mathrm{up}}, \pi_p^{\mathrm{down}})}[R(\tau)]$.

Assume now that after training with $\pi_p^{\mathrm{mix}}$, the ego policy no longer deterministically selects up, but instead plays down with non-zero probability. In this case, interactions with $\pi_p^{\mathrm{down}}$ no longer yield deterministic failure, and therefore begin to induce non-zero return variance. Consequently, partners that were previously uninformative under $\ell_{\mathrm{var}}$ may later become learnable training partners.

This illustrates an important property of UPD: partner usefulness is not static, but depends on the current ego policy. As the ego policy changes over training, the set of high-learnability partners also changes, naturally inducing an adaptive curriculum over coordination conventions.

**Large-scale generation without learnability scoring** Interestingly, this example also suggests why large-scale online partner generation alone may already induce useful curricula even without explicit learnability-based filtering. As the ego policy evolves, different randomly generated partners naturally vary in compatibility and coordination difficulty. Consequently, the distribution of effective training interactions changes over time, exposing the ego agent to progressively different coordination conventions and partner behaviours. Learnability-based selection can therefore be interpreted as a mechanism for prioritising particularly informative partners within an already adaptive partner-generation process.

# D. Additional Comparison with ROTATE and fine-tuned UPD

In Table 3 we additionally compare against ROTATE (Wang et al., 2025) and fine-tuned UPD. ROTATE uses the implementation, hyperparameters, and network architecture as provided by Wang et al. (2025) since it shows great performance in the original work and as such has been independently tuned (see Appendix E). ROTATE uses the recurrent policy proposed by Wang et al. (2025). While we otherwise compare 6 seeds, for ROTATE we report 3 seeds due to its runtime. Additionally, to show that UPD could in principle benefit from a set of per-layout hyperparameters, we also included a version of UPD in which only the curriculum parameters were fine-tuned per layout (see Table 5). Here we also report results from 3 seeds.

UPD (fine-tuned) achieves highest average returns, followed by UPD, and then ROTATE. ROTATE performs best on three layouts (CRoom, CR, FC) but sometimes error bars overlap with UPD (FC) or fine-tuned UPD (CR & FC). Our takeaway is that UPD, despite its simplicity, can compete with strong modern AHT methods. Additionally, we also see that while UPD generally works well with a shared hyperparameter configuration, it can also benefit from layout-specific tuning.

# E. Additional Training Details

## E.1. Reinforcement Learning Details

We employ independent PPO (de Witt et al., 2020; Schulman et al., 2017) for all methods in the main paper. We give an overview of all hyperparameters in Table 4, Table 6, and considered ranges in Table 7. UPD extends SFL to the partner space and therefore inherits several hyperparameters introduced by (Rutherford et al., 2024). Specifically, (1) # generated agents controls how many partners are created for learnability scoring, (2) the buffer size controls the number of agents admitted to the buffer, (3) $N$ controls the number of scoring games, (4) $R$ the rate at which the buffer is refreshed, and (5) $\rho$ determines the proportion of partners drawn from the buffer versus newly created partners during each learning iteration. For $\rho$, Rutherford et al. (2024) recommend a default value of 0.5.

We use several baselines in the paper: SP, FCP, MEP, E3T and ROTATE. We base these baselines on several open-source implementations. For FCP and E3T specifically, we built upon the open-source code of (Jha et al., 2025) while ROTATE

*Table 4.* Default hyperparameters used in our Overcooked-AI experiments.

| Category | Value |
|---|---|
| # Environments | 512 |
| Total timesteps | $5 \times 10^7$ |
| Reward shaping horizon | $3 \times 10^7$ |
| Learning rate | $1 \times 10^{-3}$ |
| Learning rate annealing | Linear |
| Seeds used | 0 - 5 |
| *PPO hyperparameters* | |
| PPO rollout length | 400 steps |
| PPO epochs | 6 |
| Minibatches per update | 8 |
| Discount factor ($\gamma$) | 0.99 |
| GAE parameter ($\lambda$) | 0.95 |
| Clipping threshold | 0.2 |
| Entropy coefficient | 0.01 |
| Value loss coefficient | 1.0 |
| Gradient norm clipping | 0.5 |
| *Architecture (shared for actor and critic)* | |
| Embedding layers | 2 |
| Actor layers | 4 |
| Critic layers | 4 |
| Fully connected layer size | 256 |
| GRU hidden size | 256 |
| Activation | Tanh |
| Layer normalisation | Enabled |
| *Partner modelling (E3T/UPD only)* | |
| Auxiliary model depth | 4 layers (size 64) |
| MOA loss coefficient | 1.0 |
| Trajectory history length | 5 steps |
| Action embedding size | 256 |
| Prediction normalisation | L2 norm |
| *UPD-specific parameters* | |
| # generated agents (SFL batch size) | 4000 |
| Buffer size ($|\mathcal{B}|$) | 512 |
| $N$ | 10 |
| $R$ | 4 |
| Sample ratio $\rho$ | 0.5 |

*Table 5.* UPD fine-tuned hyperparameters which evaluate how much UPD could benefit from layout-specific curriculum hyperparameters.

| Category | CRoom FT | AA FT | CR FT | CC FT | FC FT |
|---|---|---|---|---|---|
| Buffer size ($|\mathcal{B}|$) | 1024 | 1024 | 128 | 1024 | 128 |
| $N$ | 5 | 10 | 5 | 10 | 5 |
| $R$ | 4 | 8 | 1 | 1 | 1 |
| # sampled partners | 128 | 128 | 512 | 128 | 512 |

*Table 6.* ROTATE hyperparameters use the values reported in (Wang et al., 2025) and the respective code release. We reused their hyperparameters as they showed strong performance in the original publication.

| Category | CRoom | AA | CR | CC | FC |
|---|---|---|---|---|---|
| # Environments | 16 | 16 | 16 | 16 | 16 |
| PPO rollout length | 400 | 400 | 400 | 400 | 400 |
| Reward shaping horizon | Full | Full | Full | Full | Full |
| Discount factor ($\gamma$) | 0.99 | 0.99 | 0.99 | 0.99 | 0.99 |
| GAE parameter ($\lambda$) | 0.95 | 0.95 | 0.95 | 0.95 | 0.95 |
| Value loss coefficient | 0.5 | 0.5 | 0.5 | 0.5 | 0.5 |
| Gradient norm clipping | 1.0 | 1.0 | 1.0 | 1.0 | 1.0 |
| Seeds used | 20374 - 20376 | 20374 - 20376 | 20374 - 20376 | 20374 - 20376 | 20374 - 20376 |
| *ROTATE specific* | | | | | |
| Open Ended Iterations | 30 | 30 | 20 | 20 | 20 |
| Regret SP Weight | 3.0 | 2.0 | 2.0 | 2.0 | 2.0 |
| *Teammate PPO hyperparameters* | | | | | |
| Timesteps per Iter (T) | $6 \times 10^6$ | $6 \times 10^6$ | $1.6 \times 10^7$ | $1.6 \times 10^7$ | $1.6 \times 10^7$ |
| Learning rate (T) | $1 \times 10^{-4}$ | $1 \times 10^{-4}$ | $1 \times 10^{-3}$ | $1 \times 10^{-3}$ | $1 \times 10^{-3}$ |
| Learning rate annealing (T) | Yes | No | No | No | No |
| PPO epochs (T) | 20 | 20 | 20 | 20 | 20 |
| Minibatches per update (T) | 8 | 8 | 8 | 8 | 8 |
| Clipping threshold ($\epsilon$) (T) | 0.2 | 0.3 | 0.1 | 0.1 | 0.1 |
| Entropy coefficient (T) | 0.01 | 0.01 | 0.05 | 0.05 | 0.05 |
| *Ego PPO hyperparameters* | | | | | |
| Timesteps per Iter (Ego) | $2 \times 10^6$ | $2 \times 10^6$ | $6 \times 10^6$ | $6 \times 10^6$ | $6 \times 10^6$ |
| Learning rate (Ego) | $5 \times 10^{-5}$ | $5 \times 10^{-5}$ | $3 \times 10^{-5}$ | $5 \times 10^{-5}$ | $1 \times 10^{-5}$ |
| Learning rate annealing (Ego) | No | No | No | No | No |
| PPO epochs (Ego) | 10 | 10 | 10 | 10 | 5 |
| Minibatches per update (Ego) | 8 | 8 | 8 | 8 | 8 |
| Clipping threshold ($\epsilon$) (Ego) | 0.1 | 0.1 | 0.1 | 0.1 | 0.1 |
| Entropy coefficient (Ego) | 0.001 | 0.001 | 0.001 | 0.001 | $1 \times 10^{-4}$ |

*Table 7.* Training hyperparameters search space used for the AHT methods used in the Overcooked-AI experiments. We put the choice in **bold**. The choice is shared between the AHT methods. To decide which parameters to keep, we looked at evaluation results with a BRDiv population as a cheap proxy. Finally, for OGC experiments, we reused the above-found hyperparameters in bold and only added more environments (1024), steps ($1 \times 10^9$), and actor and critic layers (1 extra; see text). If $CV^2$ is used we pick the stability constant as $\delta = 10^{-8}$. For LBF we used the same hyperparameters as in Overcooked-AI except for the values mentioned in the main paper.

| Category | Hyperparameter Range |
|---|---|
| Total timesteps | $5 \times 10^7$ |
| Reward shaping | $3 \times 10^7$ |
| Learning rate | $\mathbf{1 \times 10^{-3}}, 3 \times 10^{-4}, 1 \times 10^{-4}$ |
| PPO epochs | 4, **6** |
| PPO minibatches | 4, 6, **8** |
| Layernorm? | **True**, False |
| *UPD-specific parameters* | |
| Buffer size ($|\mathcal{B}|$) | 128, 256, **512** |
| $N$ | 3, **10** |
| Buffer refresh freq. | 1, 2, **4** |

uses (Wang et al., 2025). We adopt MEP from FCP and Zhao et al. (2023). SP-IPPO uses the shared PPO hyperparameters, except for a value function coefficient of 0.5, learning rate of $2.5 \times 10^{-4}$, and 4 PPO epochs/minibatches. We found the default configuration to be unstable in standard self-play and perform substantially worse in AHT evaluation (average artificial-partner performance: 32.4 vs. 41.4), so we report the stronger configuration throughout.

Following standard practice, the training populations for FCP and MEP use MLP agents. They generally built on the same hyperparameters as the IPPO-SP baseline and otherwise follow Table 4. We train 16 seeds each and extract 3 checkpoints per seed for a population size of 48. We use checkpoints at initialization, 50% of final reward, and final performance, following standard protocol. Our FCP population trains entirely in parallel by dividing a total of 1024 training environments and $2 \times 10^8$ training steps between them. The resulting FCP populations achieve high self-play returns. MEP build on FCP and uses a sequential training setup where one agent is trained for one PPO update cycle while the remaining agents are used to calculate the entropy shaping term. MEP generally uses an entropy coefficient of $\alpha = 0.01$. For MEP we found that the population generation hyperparameters sometimes needed to be adjusted per layout to ensure stronger results. We therefore tuned MEP population parameters per layout. For CC and FC we used 10 PPO epochs, 64 minibatches, and PPO entropy coefficient 0.03. For FC, where higher population-entropy coefficients led to unstable population learning, we used $\alpha = 0.001$ and clipping threshold 0.1. We generally found an interaction effect between population size and $\alpha$ where larger populations generally require careful tuning to converge stably.

ROTATE used the published hyperparameters and code as described in (Wang et al., 2025). We use these settings because ROTATE was tuned per Overcooked-AI layout in the original work and shows strong performance over strong baselines and evaluation partners that overlap with those considered here. Since their default random seed is 20374 we use 20374, 20375 and 20376 as seeds for training. We show the full hyperparameters in Table 6. The total number of environment interactions ROTATE learns from is given by the number of timesteps per iteration for both the teammate and ego multiplied by the number of open ended iterations. Using the released implementation and hyperparameters, this corresponds to $6.0 \times 10^7$ ego steps and $2.4 \times 10^8$ total steps for CRoom/AA, and $1.2 \times 10^8$ ego steps and $4.4 \times 10^8$ total steps for CR/CC/FC.

Experiments on the OGC used the exact same hyperparameters with two differences: we use an additional actor and critic layer (5 each) and train in 1,024 environments for $1 \times 10^9$ total timesteps. SFL based methods for OGC (SFLE3T, JUPD) use 8192 generated levels, use a rollout factor of $N = 5$, $\rho = 1.0$ and larger buffer sizes (4096 and $16,384$). JUPD requires a larger buffer since it generates 12 partners per generated level and scores the pair. Additionally, JUPD uses $R = 2$. Training curves are shown in Figure 15, and SFLE3T achieves the highest training performance while still underperforming JUPD in partner-level generalisation.

**Partner Populations**    We reuse the implementation of Wang et al. (2025) for generating artificial evaluation partners. BRDiv partner populations were tuned to be strong but incompatible per layout which is characterised by high self-play and low cross-play returns. Our hardcoded partners act independently or mechanically perform only subtasks of the overall task. For FC (again, because of its restrictive dynamics) we only use BRDiv and the independent partner. Since the independent partner cannot solve the FC task alone, it is programmed to have a 60% chance of dropping an onion or plate onto a counter nearby. This matches prior work (Wang et al., 2025).

### E.2. Neural Network Architecture

We employ a recurrent actor-critic architecture for all methods in Table 1. The model comprises a shared encoder, a recurrent processing module, and separate heads for policy and value estimation.

Each agent's observation is passed through a feed-forward encoder consisting of a linear layer followed by a configurable number of fully connected layers (default: two layers). Each layer contains 256 hidden units with either ReLU or Tanh activations, optionally followed by layer normalisation. The resulting representation is used as input to the recurrent module.

Temporal dependencies are captured using a Gated Recurrent Unit (GRU) (Cho et al., 2014) with a hidden state size of 256. The recurrent hidden state is reset at environment terminal states. The GRU output serves as a temporal embedding and is shared by both the actor and critic heads.

The policy head processes the recurrent embedding through four fully connected layers with 256 units each and non-linear activations. A final linear layer outputs unnormalised logits over discrete actions, defining a categorical action distribution. The value head also uses the recurrent embedding as input and applies four fully connected layers with 256 units each, followed by a scalar output representing the state value estimate.

We employ the *other agent modelling network* proposed by E3T. This auxiliary module models the behaviour of its teammate. Each agent receives the past five state-action pairs of the other agent. Observations and actions are embedded and passed through a four-layer multilayer perceptron (64 units per layer) to predict the teammate's next action distribution. The prediction is L2-normalised and concatenated with the agent's own embedding before being fed into the policy head. This auxiliary loss is optimised using a cross-entropy objective and weighted by a tunable coefficient. This is consistent with the original formulation (Yan et al., 2023).

For ROTATE, we used their architecture as described in (Wang et al., 2025) to match their reported hyperparameters.

## F. Observations on Computational Requirements of UPD

We compared the computational requirements of population-based approaches and UPD. Below, under conservative assumptions, we find that for our settings and any reasonably large population size ($n \geq 3$) UPD can incur lower computational requirements. In practice, Jax-based implementations make UPD, E3T, and best response (MEP and FCP) training runs comparably fast; however, population pretraining dominates the runtime for both MEP and FCP, as it scales with the population size. In our experiments, UPD runs at close to self-play speed, given modern Jax-based environment simulators where policy gradient updates dominate environment steps in cost.

In the following, we show how our curriculum compares in terms of computational requirements to common population-based approaches. We assume that agents are optimised using PPO (Schulman et al., 2017), but similar analysis is possible for different RL algorithms.

End-to-end training with PPO has two major costs: the cost of computing an environment transition $C_{\text{env}}$ and the cost of updating the policy using the PPO loss via back-propagation $C_{\text{PPO}}$. Let $E$ be the number of parallel environments and $H$ the number of *training* steps per environment per loop, so the total training batch per loop is $N_{\text{steps}} := EH$. In UPD, partner *scoring* uses $N$ evaluation rollouts of length $L$ steps per candidate; $L$ need not equal $H$.

With a population of size $n = |P|$, training the population and then a best response incurs

$$\text{Cost}_{2\text{ Stage}} \approx (n+1)N_{\text{loops}} \left( N_{\text{steps}} C_{\text{env}} + C_{\text{PPO}} \right). \tag{16}$$

In contrast, UPD removes population training and adds a partner-scoring term: every $R$ PPO loops, UPD fully refreshes a top-$|\mathcal{B}|$ partner buffer by evaluating $N$ rollouts of length $L$ for each candidate. In the following, let $K$ be the number of generated partners. Since partner generation is practically computationally free as it requires only sampling a few parameters, this adds

$$\text{EnvSteps}_{\text{UPD}} \approx N_{\text{loops}} \frac{K\,N\,L}{R}. \tag{17}$$

However, since candidate partners differ only by simple perturbations on the logits, we evaluate the $K$ candidates in parallel (using $K$ simulators), so the overhead per PPO loop scales with

$$\text{Overhead}_{\text{UPD}} \approx \frac{N\,L}{R}, \tag{18}$$

rather than $KNL/R$. Similarly, under vectorised execution the wall-clock rollout cost scales primarily with the rollout horizon $H$, rather than the total number of environment transitions $EH$.

Thus, the total per-loop wall-clock cost for UPD is

$$\text{Cost}_{\text{UPD}} \approx N_{\text{loops}} \left( HC_{\text{env}} + C_{\text{PPO}} + \frac{NL}{R} C_{\text{env}} \right), \tag{19}$$

i.e., the standard PPO loop plus a tunable scoring term.

Population methods are more expensive than UPD when

$$(n+1)\left( HC_{\text{env}} + C_{\text{PPO}} \right) > HC_{\text{env}} + C_{\text{PPO}} + \frac{NL}{R} C_{\text{env}} \tag{20}$$

Solving for $n$ yields the break-even population size

$$n^{\star} = \frac{\frac{NL}{R}}{H + \frac{C_{\text{PPO}}}{C_{\text{env}}}}. \tag{21}$$

In the worst-case where environment transition costs dominate PPO update costs, e.g. the environment-dominated limit $C_{\text{PPO}}/C_{\text{env}} \to 0$,

$$n^\star = \frac{NL/R}{H}. \tag{22}$$

To be cheaper than any non-trivial population ($n \geq 1$), *it suffices that* $\frac{NL}{R} \leq H$ (equivalently, $n^\star \leq 1$). More generally, to undercut a given population size $n$, *it suffices that* $\frac{NL}{R} \leq n H$. In practice, choosing modest $N$, short $L$, and staggered refresh $R$ keeps $\frac{NL}{R}$ small.

In our Overcooked-AI runs we use $E = 512$, $H = 400$ (so $N_{\text{steps}} = 204,800$), $N = 10$, $L = 400$, $R = 4$, and $K = 4000$, giving an amortised scoring budget $N L/R = 1000$. Hence

$$n^\star = \frac{\frac{NL}{R}}{H + \frac{C_{\text{PPO}}}{C_{\text{env}}}} = \frac{1000}{400 + \frac{C_{\text{PPO}}}{C_{\text{env}}}}. \tag{23}$$

Two implications follow. First, if $C_{\text{PPO}}/C_{\text{env}} > 600$, then $n^\star < 1$, i.e., even a single-partner population already exceeds UPD's wall-clock overhead. Second, in the simulation-dominated extreme $C_{\text{PPO}}/C_{\text{env}} \to 0$, we obtain $n^\star = 1000/400 = 2.5$, so any practical population with $n \geq 3$ is more expensive than UPD. Equivalently, the per-loop UPD overhead is a fixed $1000\,C_{\text{env}}$, which as a fraction of the PPO update cost is $1000/(C_{\text{PPO}}/C_{\text{env}})$ (vanishing when $C_{\text{PPO}} \gg C_{\text{env}}$). Note that, (1) in this work we are in the PPO-dominated regime since we use Jax-based simulators that run at $> 100000$ steps/second and (2) that costs are even higher for methods that revolve around computing interactions between members of the population.

The above additionally assumes that different population members are not themselves trained in parallel. If the $n$ population members can themselves be trained fully in parallel (which is not true for most methods), the wall-clock cost of the population pretraining stage no longer scales with $n$. In this best-case parallel setting, a two-stage method still requires two sequential phases: population pretraining followed by best-response training. Its cost is therefore approximately

$$\text{Cost}_{\text{2 Stage, parallel}} \approx 2N_{\text{loops}} \left( HC_{\text{env}} + C_{\text{PPO}} \right). \tag{24}$$

UPD remains cheaper when

$$HC_{\text{env}} + C_{\text{PPO}} > \frac{NL}{R}C_{\text{env}}, \tag{25}$$

or equivalently

$$\frac{C_{\text{PPO}}}{C_{\text{env}}} > \frac{NL}{R} - H. \tag{26}$$

With our values $N = 10$, $L = 400$, $R = 4$, and $H = 400$, this threshold is

$$\frac{C_{\text{PPO}}}{C_{\text{env}}} > 600. \tag{27}$$

This condition is realistic and likely conservative in our setting: our environments are lightweight Jax simulators, while $C_{\text{PPO}}$ includes the full PPO optimisation phase over all minibatches and epochs. Empirically, UPD runs close to self-play/E3T speed, whereas population-based methods remain slower due to the additional pretraining stage and, for methods such as MEP, additional population-interaction costs.

## G. Additional Environment Details

### G.1. Level-Based Foraging

As described in the main text, we employ Level-Based Foraging (LBF) (Albrecht & Ramamoorthy, 2013) to first compare E3T to UPD. In our work LBF uses a grid of $7 \times 7$, three foods and two agents. Both agents observe the full environment. To collect a food, both agents need to use the `load` action on a food at the same time. The actions are: `up`, `down`, `left`, `right`, `no-op`, and `load`. On reset the locations of foods and agents are randomised. We use the Jax version of the environment offered by Bonnet et al. (2024) with improvements made by Wang et al. (2025) and the partner generation pipeline of Wang et al. (2025) for evaluation partners.

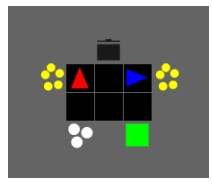 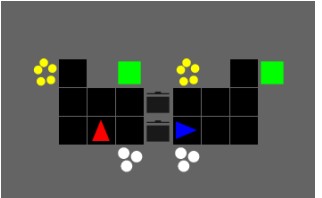 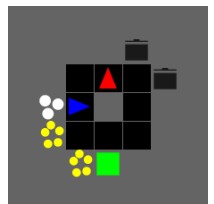 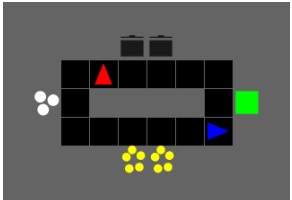 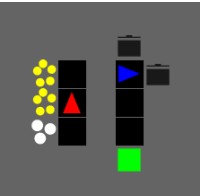

*Figure 9.* We redraw the five evaluation layouts introduced by Carroll et al. (2019) using the JaxMARL visualisation pipeline (Rutherford et al., 2023). From left to right: CRoom, AA, CR, CC, and FC.

### G.2. Overcooked-AI

Overcooked-AI is a multi-agent coordination benchmark based on the Overcooked video game, originally proposed by Carroll et al. (2019). It features two-player cooking tasks requiring temporal coordination and spatial reasoning. Agents must pick up ingredients, cook them in pots, and serve dishes. The action space is discrete, consisting of six actions: move up, down, left, right, interact, and stay. We use the JaxMARL version (Rutherford et al., 2023) with improvements provided by Wang et al. (2025).

We use the five standard layouts commonly used in ad-hoc teamwork literature (also see Figure 9): Cramped Room (CRoom), Asymmetric Advantages (AA), Coordination Ring (CR), Counter Circuit (CC), and Forced Coordination (FC). We use the implementation provided by JaxMARL and keep all reward settings at their defaults (sparse reward for serving, shaped reward for intermediate actions where indicated). During the shaped reward phase, agents receive a reward of 3 for placing an ingredient into a pot, 3 for picking up a plate while a soup is cooking, and 5 for picking up a ready soup.

### G.3. The Overcooked Generalisation Challenge

The Overcooked Generalisation Challenge (OGC) (Ruhdorfer et al., 2025b) extends Overcooked-AI to assess zero-shot generalisation across both partner and level distributions. Instead of training on fixed layouts, the environment includes a procedural level generator that produces randomised kitchens with varying topology and difficulty.

The challenge is particularly demanding because many generated layouts are unsolvable or require nontrivial conventions to coordinate efficiently.

In our work, we use a $5 \times 5$ version of the OGC to reduce computational cost while preserving task diversity. This generator randomly samples kitchen structure (walls, counters, item placement) and goal configurations (e.g., number of onions per soup). The generator first generates a layout with walls at the border and then randomly samples a wall budget between 1 and 10. With this budget, the system then places walls randomly. The generator also randomly adds a dividing wall or additional walls at the sides in order to narrow the layout. After this, the system places items on walls. Lastly, both agents are placed on a free tile.

We evaluate performance on a fixed subset of small layouts used in the OGC benchmark and compare agents based on average return when paired with held-out agents across these generated environments.

## H. Training Curves

We display the received training returns for all Overcooked-AI methods in Figures 10 (SP), 11 (FCP), 12 (MEP), 13 (E3T) and 14 (UPD). Note that these returns must be interpreted with caution. For FCP and MEP, they show the returns with their respective populations. For UPD and E3T, they show the received training returns based on the generated partner. All reported configurations converge stably. Also note that many methods achieve higher training returns compared to UPD, but, as has been established in the literature (Carroll et al., 2019), this is not necessarily predictive of test returns with unknown partners.

In Figure 15, we display training returns for all four OGC methods. We see all methods converging. Only CEC appears to experience a drop in performance at the end. Again, note that training performance is not predictive of test-time performance: Methods that sample levels by learnability (SFLE3T, JUPD) might sample harder levels than methods that do not (DR, CEC), and both SFLE3T as well as JUPD do not play in a self-play setting.

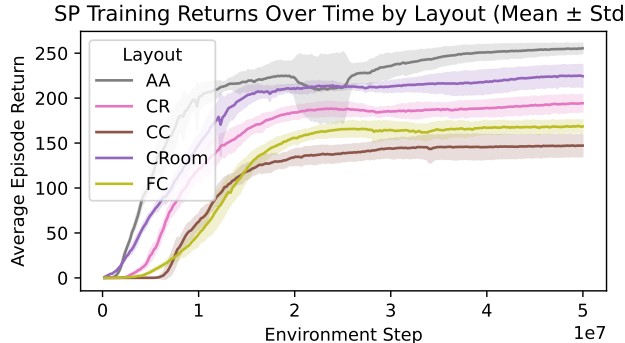

*Figure 10.* SP training curves. We average over 6 seeds and show standard deviation.

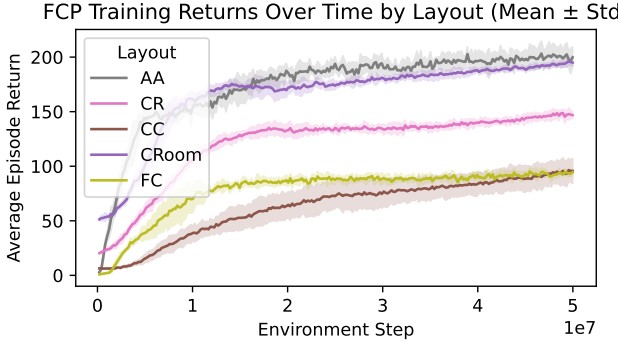

*Figure 11.* FCP training curves. We average over 6 seeds and show standard deviation.

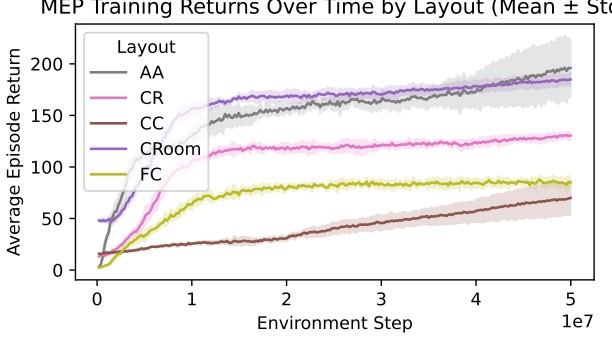

*Figure 12.* MEP training curves. We average over 6 seeds and show standard deviation.

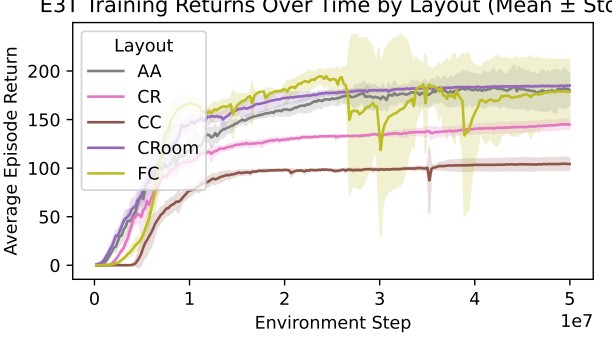

*Figure 13.* E3T training curves. We average over 6 seeds and show standard deviation.

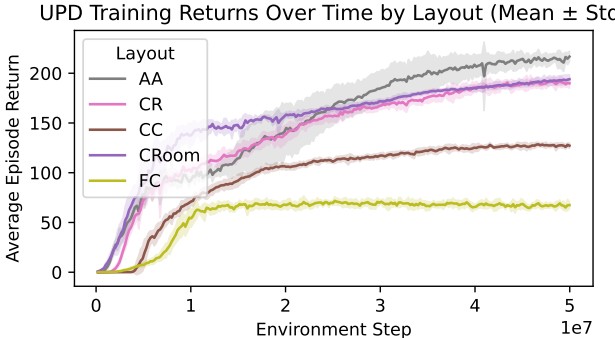

*Figure 14.* UPD training curves. We average over 6 seeds and show standard deviation.

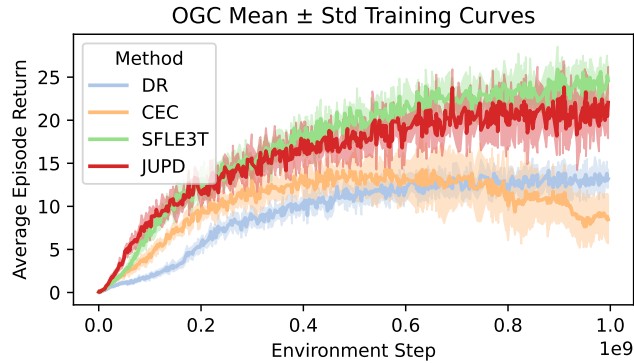

*Figure 15.* OGC training curves. We average over 6 seeds and show standard deviation for all methods.

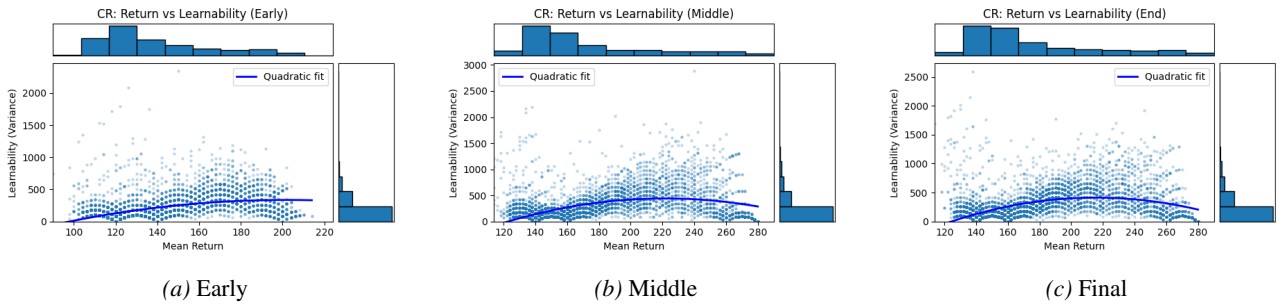

*(a)* Early  *(b)* Middle  *(c)* Final

*Figure 16.* Learnability vs. return over training in **Coordination Ring**.

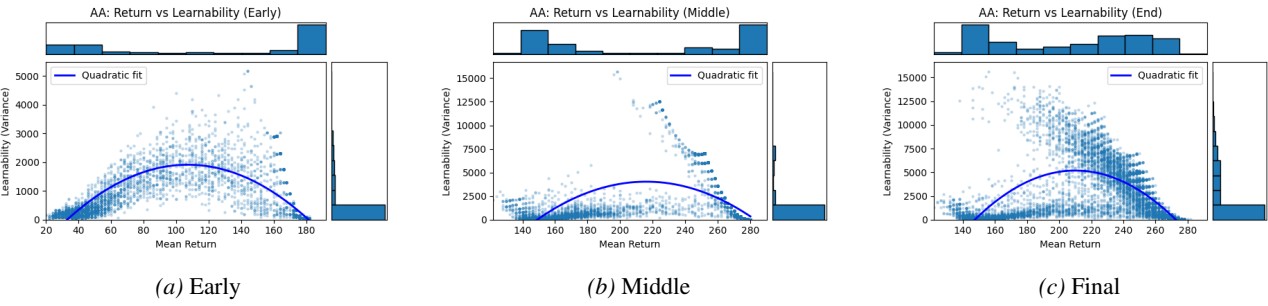

*(a)* Early  *(b)* Middle  *(c)* Final

*Figure 17.* Learnability vs. return over training in **Asymmetric Advantages**.

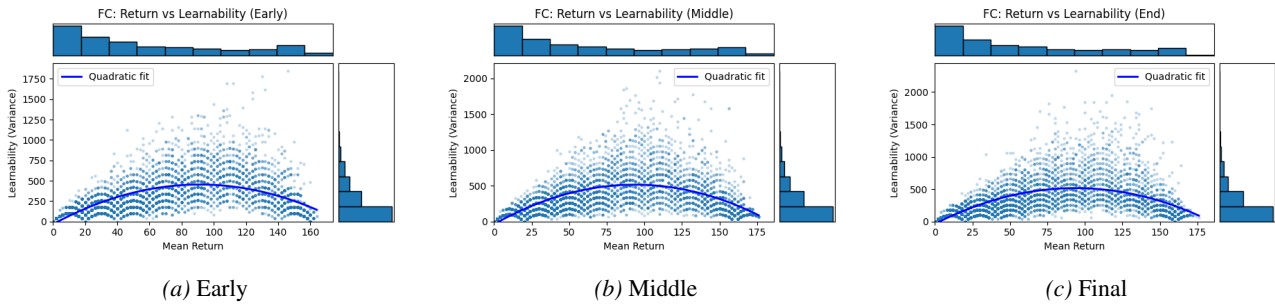

*(a)* Early        *(b)* Middle        *(c)* Final

*Figure 18.* Learnability vs. return in **Forced Coordination**.

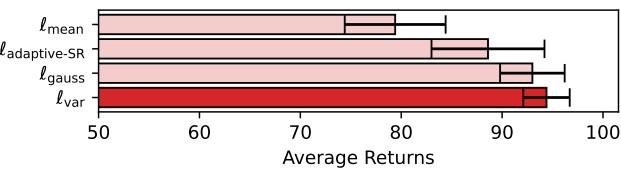

*Figure 19.* Performance of multiple learnability functions against the evaluation population averaged over layouts. Here $\ell_{\text{mean}}$ selects partners based on their mean return, $\ell_{\text{gauss}}$ weights partners down that are far from the average global return (Monette et al., 2025) and adaptive success-rate (SR) ranks partners based on whether they exceed median performance. Simple $\ell_{\text{var}}$ performs similar or better than more complicated measures. Additionally, UPD is robust to the learnability function as long as the function is reasonable. Researching other forms of partner scoring might be valuable future work in the context of UPD.

## I. Partner Curriculum Dynamics

To better understand how UPD identifies effective training partners throughout learning in Overcooked-AI, we repeat our analysis from the experimental section and visualise learnability scores at different points across training. For three respective layouts (CR, AA, and FC), we evaluate a trained ego agent at three points during training (at 2/5, 4/5, and the end) by rolling out with 8,192 randomly generated partners (see Figures 16, 17, 18). We select these three layouts for the same reason as in the main paper: they are representative; both CRoom and CC behave very similarly to the CR case. We compute learnability using the same variance-based metric as during training and plot it against the corresponding mean return. Across all settings, we observe that learnability does not concentrate on the highest- or lowest-returning partners, nor does it peak where most partners lie. Instead, learnability is high for partners with intermediate difficulty. We observe that for most levels, the generated partners fall into low-return and low-learnability buckets. This suggests that our hypothesis – that not all partners are optimal for training – is correct: Most partners score low on the learnability metric. Instead, UPD ranks partners from middle-return buckets highest, which contain relatively few partners.

Finally, unlike prior work on level-space curricula with binary outcomes (Rutherford et al., 2024), we observe that agents in our setting rarely score zero reward with any partner. Due to this and the continuous reward structure, we find that our learnability analysis plot conveys information not only through its shape but also via its rightward shift over time. For instance, in Figure 17, the lowest-scoring partner improves from around 20 to a mean return of 120. This shift suggests that the ego agent's capabilities expand over time and that UPD adapts by sampling increasingly competent partners. In doing so, UPD not only identifies informative partners but also tracks the agent's learning progress, adjusting the curriculum accordingly. A notable exception arises in tasks with strong interdependence, such as FC (see Figure 18).

## J. Additional Results in Overcooked

### J.1. Do alternative learnability functions improve results?

We examined how different learnability functions affect partner selection and training performance. Specifically, we asked: (1) Does our proposed variance-based score $\ell_{\text{var}}$ induce effective curricula? and (2) How sensitive is UPD to the choice of learnability function? We evaluated four functions across the five Overcooked-AI layouts from RQ2 using the following comparisons: (1) $\ell_{\text{mean}} = \mathbb{E}_{\tau \sim (\pi, \pi_p)}[R(\tau)]$, selects partners with higher expected returns. (2) We adapted the success-rate-based learnability of Rutherford et al. (2024) to continuous rewards by thresholding returns at the median, i.e.

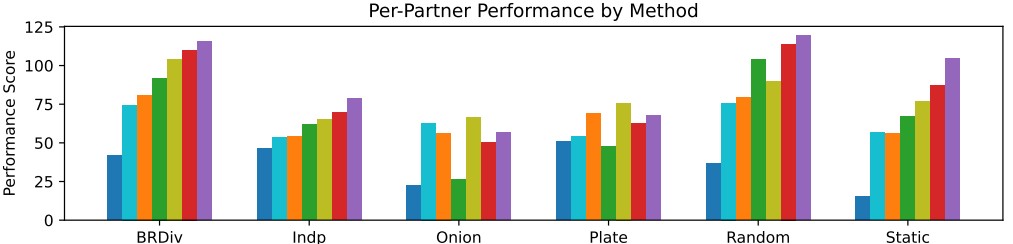
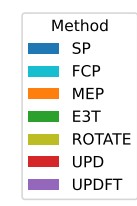

*Figure 20.* Per-Partner results averaged over layouts in Overcooked-AI. UPD outperforms other methods with most partners. Some methods show similar or even slightly better performance with the Onion and the Plate agent. However, these methods are not consistently better than UPD.

using an estimated pseudo-success rate:

$$\ell_{\text{adaptive-SR}} = p \cdot (1 - p), \tag{28}$$

$$\text{where } p \text{ is the fraction of rollout returns exceeding the global median return.} \tag{29}$$

(3) The Gauss-weighted formulation ($\ell_{\text{gauss}}$) of (Monette et al., 2025). (4) Our $\ell_{\text{var}}$. Together, these cover naive reward-driven and other selection strategies identified in related works.

As shown in Figure 19, our $\ell_{\text{var}}$ achieves the highest performance. Notably, UPD remains robust across alternative functions – as long as they avoid pathological cases (e.g., self-play). Contrary to prior findings (Monette et al., 2025), we observe that more complex learnability functions offer no clear benefit for UPD.

Our experiments showed that UPD is effective at training agents for AHT, and is robust under different formulations of learnability as long as they do not collapse to the self-play setting. Additionally, we demonstrated that UPD can be integrated into standard UED, enabling fully unsupervised curricula over both environment and partner distributions. While we explored several learnability functions, many alternatives remain. One natural baseline is to prioritise the hardest partners – those with the lowest average return – as explored in prior work (Zhao et al., 2023; Li et al., 2023b; You et al., 2025). However, both theoretically and in preliminary experiments, we find this approach leads to adversarial dynamics. Specifically, in Overcooked-AI, it heavily favours near-random partners (i.e., $\epsilon \to 1$), while in the OGC, it favours unsolvable levels. In both cases, learning stagnates. This is because our setting involves an open-ended partner generator, unlike prior work, which relies on pre-trained or bounded populations that implicitly cap adversarial behaviour. In such constrained settings, prioritising "hard" examples remains within the space of feasible coordination, whereas in (J)UPD, unconstrained difficulty selection can collapse the training signal entirely.

### J.1.1. PER PARTNER RESULTS

In Figure 20, we display additional results for how UPD performs with different partners. On average and with most partners, UPD performs best.

### J.1.2. CURRICULUM DYNAMICS OF UPD ABLATIONS

To better understand the role of the bias mechanism in the induced curriculum, we additionally analyse the partner-selection dynamics of UPD w/o bias. Figure 21 shows the average selected mixing coefficient $\epsilon$ throughout training across layouts. Overall, we observe curriculum dynamics similar to those of full UPD (Figure 4). The main difference occurs in CRoom, where UPD w/o bias converges to a lower average $\epsilon$ than full UPD.

In contrast, UPD w/o $\ell$ fluctuates around a mean $\epsilon$ of $0.5$ across all layouts, as expected. Additionally, we examined the evolution of action biases for UPD w/o $\ell$. When partners are sampled uniformly without curriculum filtering, the induced action biases remain approximately constant throughout training and fluctuate around the random-policy prior. This suggests that the convention-breaking dynamics observed in Figure 6 emerge from the interaction between partner generation and learnability-based selection, rather than from the stochastic partner generator alone.

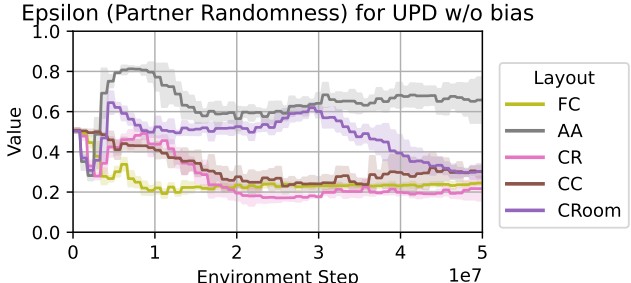

*Figure 21.* We show the same analysis as in Figure 4, but for UPD w/o bias. We observe similar curriculum dynamics overall: only on CRoom does UPD w/o bias converge to a noticeably lower average $\epsilon$ compared to full UPD.

## K. Additional User Study Details

We employed a within-subjects design: each participant interacted with all 4 agents (SP, MEP, E3T, UPD) across 3 Overcooked layouts (AA, CR, CC). Each agent-layout pair was played once, totalling 36 episodes per participant. Agent and layout order were randomised per user. Each full study lasted between 18 and 35 minutes, and participants completed a short tutorial prior to the main study. Games are 200 environment steps (20 points per delivery), matching prior human evaluation protocols (Jha et al., 2025).

The study was conducted via a web-based interface using the `NiceWebRL` framework[2]. We recruited 12 participants (5 female; aged 22-31) for the user study. The study was approved by our institutional ethics review board, and all participants gave informed consent. Compensation was provided per institutional guidelines. Participants controlled their character using the keyboard (arrow keys, space and the S key). Participants were not informed about the agent identities to avoid bias and were also unknown to the experimenter during the experiment (double-blind). After each agent interaction, participants rated the agent using seven 5-point Likert-scale questions (strongly disagree, disagree, neutral, agree, strongly agree). The questions were:

1. I enjoyed playing with the agent.

2. I felt that the agent's ability to coordinate with me was: (very poor, poor, neutral, good, very good)

3. The agent adapted to me when making decisions.

4. The agent frequently got in my way. *(negative)*

5. The agent was consistent in its actions.

6. The agent's actions were human-like.

7. The agent's behaviour was frustrating. *(negative)*

We additionally allowed users to give free-form feedback at the end of the study. Responses were numerically mapped from 1 to 5. Negative-valence questions were inverted before aggregation for our analysis on the overall subjective preference. To assess internal consistency of the question responses, we computed Cronbach's $\alpha = 0.916$, suggesting strong reliability. This justifies aggregating scores across questions to produce a single subjective preference score per agent.

We ran one-sided Wilcoxon signed-rank tests comparing UPD to each baseline using the hypothesis that UPD > baselines. To control for multiple comparisons, we applied Holm-Bonferroni correction within each question. We also tested the aggregated preference scores using the same procedure. Performance (reward) was analysed via one-sided paired t-tests with Holm correction. We provide bar plots for each survey question, showing Likert response distributions per agent, available in Figure 22.

---

[2] https://github.com/KempnerInstitute/nicewebrl

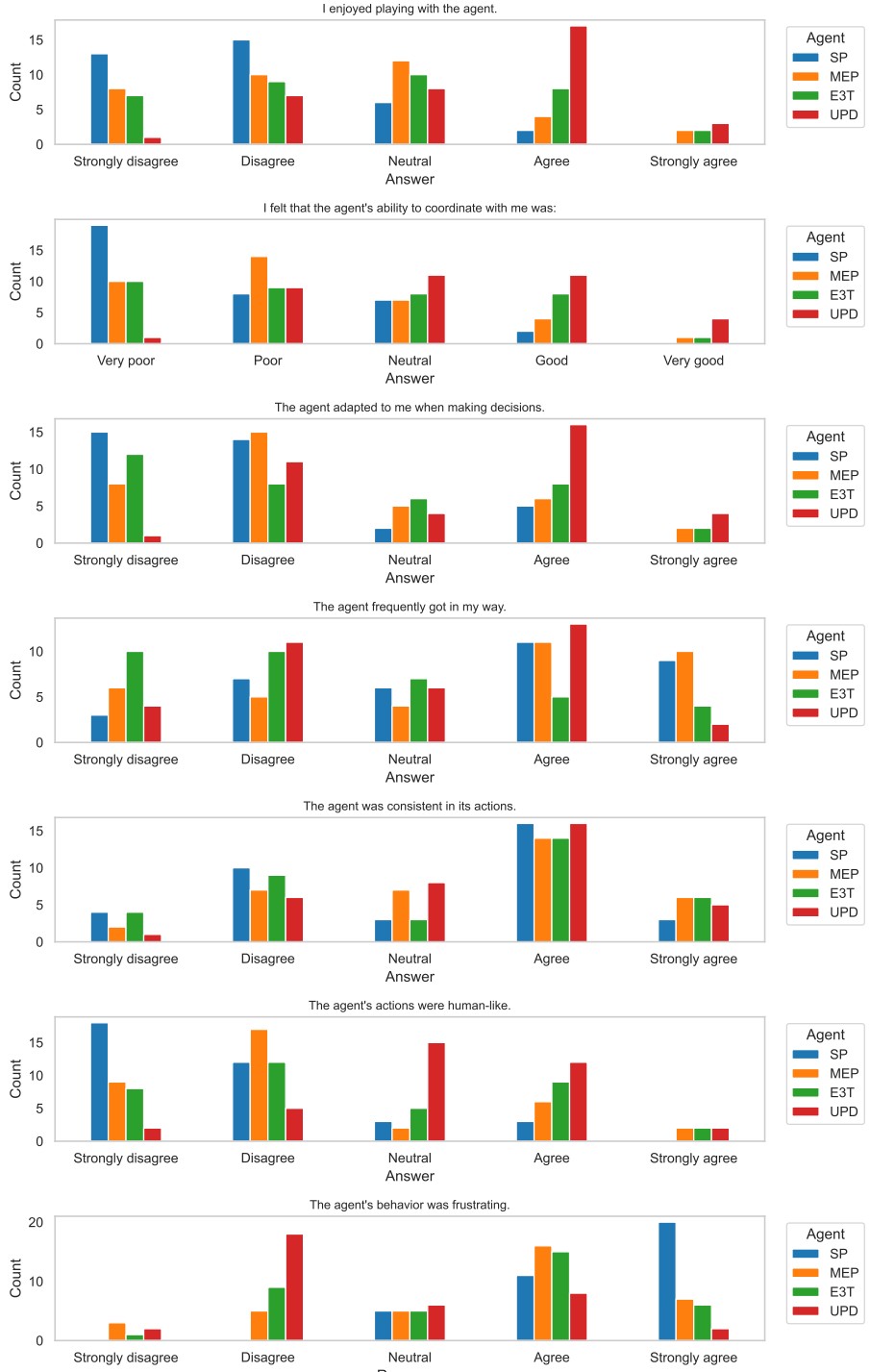

*Figure 22.* Distribution of human ratings for each survey question across all agents. Each bar represents the number of responses given to each Likert item (x-axis), with colors indicating the agent. Questions are grouped vertically and include both subjective impressions (e.g., enjoyment, consistency) and collaboration quality (e.g., coordination, frustration).

