# OpenReview forum: "Unsupervised Partner Design Enables Robust Ad-hoc Teamwork"
_ICML.cc/2026/Conference — ICML 2026 spotlight_

### Official Review · Reviewer_ZDAM · 2026-03-01

**Soundness:** 4
**Presentation:** 4
**Significance:** 3
**Originality:** 3
**Overall Recommendation:** 6
**Confidence:** 4

**Summary:**

# Summary:

## Problem:

Ad-hoc teamwork (AHT) is a core requirement for cooperative AI. However, training a partner policy without assumptions about their deployment partners makes learning coordination strategies difficult. One way in which previous approaches has addressed this difficulty is by relying on a large population of diverse partner policies, but this is costly to maintain and tune for diversity at scale that enables zero-shot coordination.

Moreover, besides generalisation across partners, it is important for the trained agent to also be able to generalise to some extent across environment parameters. Previous work has shown that using adaptive curricula over those parameters, i.e. over environment design, enables significant generalisation gain.

Thus, by analogy, the paper investigates whether partner-design mechanisms can lead to robust AHT, without the cost of maintaining an explicit population of policies, and whether such mechanisms are sufficient for joint zero-shot coordination and environment generalisation.

## Contributions:

In the context of human-AI collaboration in ad-hoc teamwork (AHT), the paper proposes Unsupervised Partner Design (UPD) , a population-free MARL approach that trains and selects partner policies based on a learnability criterion. Those partner policies are used to train a zero-shot coordination-capable ‘ego’ policy. Thus, intuitively, UPD trains and selects partner policies to lie near the learning frontier of the ‘ego’ policy.

UPD improves by ~20% (absolute) the state-of-the-art on Level-Based Foraging and Overcooked-AI benchmarks.

Moreover, the paper shows that JUPD, extension of UPD to joint curriculum over partner and task levels, is not only able to perform well with unknown deployment partners but also with relatively-unknown environments (procedurally-generated levels), as tested on the Overcooked Generalisation Challenge (OGC) where it achieves a new state-of-the-art.

**Compliance With Llm Reviewing Policy:**

Affirmed.

**Final Justification:**

# Post-Rebuttal Update:
Raising my overall score to 6 and soundness score to 4, as all my flagged weaknesses have been addressed by the rebuttal.

**Key Questions For Authors:**

Cf. Strenghts and Weaknesses above.

**Limitations:**

# Limitations:

L1: The paper fails to discuss time and space complexity limitations (cf. WQ5).

**Strengths And Weaknesses:**

# Strengths & Weaknesses:

## Quality:

SQ1: The introduction is well-written and provides an helpful big-picture viewpoint that should ease the reading experience of a diverse audience.

WQ2: As Section 4.4 is an important discussion, I think it would improve the quality of the paper if some experimental evidence were provided as well. Thus, I encourage the authors to add some experimental (and/or formal) evidence of their claim that ‘UPD will […] [discourage] overfitting to a convention while also maximising expected policy improvement’.

SQ3: I appreciate the use of a sweeping of $\epsilon$ to present the results around Figure 3, aiming for a fair comparison with previous work that was not evaluated on the current benchmark.

WQ4: I think adding ‘UPD w/o bias’ to Figure 4, in some selected layouts, would increase the quality of the current paper and allow a discussion of the extent to which the low-level biases interact in practice with the partner curriculum.

WQ5: In Algorithm 1, lines 5 to 10 detail how the partner policy candidates are evaluated in terms of learnability. As the paper claims to be ‘lightweight’ compared population-based approaches, I think the paper would improve by providing some quantitative evidence in the form of the time and space complexities of those few lines in Algorithm 1, and compare them to the time and space complexities of the compared population-based approaches (e.g. ), as well as to E3T to clearly highlight the trade-off involved. Indeed, as far as I understand, while E3T requires a fixed-but-sensitive $\epsilon$ parameterisation, I expect its time and space complexities to be far lower than those UPD, and this could be flagged as a limitation (if I am not mistaken?).

(UPD still achieves better performance than E3T, but, in some cases, having suboptimal performance of E3T without the (suspected) time and space complexity costs of UPD might sometime be of value.)

## Clarity:

SC1: The last paragraph of Section 3.2 is very helpful to understand the subsequent sections. The paper is overall very-well written.

## Originality:

SO1: The paper is well-situated in the literature.

SO2: While extending E3T from fixed $\epsilon$ to a distribution, here starting in $\mathcal{U}([0,1])$ would have only been considered incremental innovation, I think that the addition of a curriculum over partners that relies on the well-motivated connection between learnability and expected policy improvement (from Foster at al., 2025) makes the UPD method a radical innovation and well-worth sharing in ICML.

## Significance:

SS1: As (J)UPD shows SotA results in AHT and in joint ad-hoc teamwork and environment generalisation, I think this paper is very significant to the ICML community.

SS2: With mean+/-std. being reported on all experiments, all results clearly show statistical-significance compared to previous art.

SS3: I appreciate the statistical-significance tests over the human evaluation survey results. However, I think the paper could be improve by clarifying the motivation behind choosing these specific tests (and corrections) and clarifying the nature of each of the tested data.

---

> ### Author Rebuttal · Authors · 2026-03-29
>
> Dear ZDAM,
>
> Thank you for your detailed and constructive review. We are glad that you find the paper well written and the empirical evaluation strong, and we appreciate your thoughtful suggestions.
>
> ## WQ2: Empirical or theoretical evidence for 4.4
>
> Section 4.4 is intended to provide intuition for how learnability-based partner selection discourages convergence to fixed conventions while maintaining learning progress.
> Empirically, Fig. 6 shows systematic reversals in action biases over training, indicating that the agent does not settle on a single convention but continues to adapt. At the same time, Fig. 5 shows that the curriculum concentrates on partners of intermediate difficulty, which are known to maximise learning progress.
> Together, these results support the claim that UPD avoids overfitting to a fixed convention while maximising expected policy improvement by selecting partners of high learnability. We agree that this connection can be made clearer and will strengthen this discussion in the revision.
>
> ## WQ4: ‘UPD w/o bias’ to Figure 4
>
> We have conducted this analysis and will include it in the appendix. Preliminary results show that removing bias slightly shifts the $\epsilon$ distribution (e.g., towards lower values in CR and CRoom), while preserving the overall curriculum dynamics.
>
> ## WQ5: Space and Time Complexity
>
> We will include a more prominent discussion in the main paper. Appendix D of the main paper analyzes the complexity. The key tradeoff is: UPD adds rollout-based scoring overhead but removes population pretraining which both adds additional rollouts and gradient updates.
>
> Under our Overcooked settings, the scoring term is modest. Appendix D shows that under conservative assumptions UPD is cheaper than population methods with population size above 3.
>
> In wall-clock terms on our cluster, E3T takes ~38 min, UPD ~51 min, while FCP with population size 48 requires roughly 281 min including population pretraining (16 self-play agents + best response). In the response to reviewer e6kW we compare with ROTATE (a recent population extension method) which runs for multiple hours (3h30 to ~7h). Memory usage for UPD is comparable to E3T, while population-based methods scale with population size.
>
> Finally, we want to note that UPDs runtime can be controlled by e.g. the refresh rate $R$ to the available compute budget. We have performed an additional hyperparameter study and found that UPD still works well with fewer refreshes while further reducing run time.
>
> Overall, UPD introduces moderate overhead relative to E3T and remains significantly more efficient than population-based approaches while also adding substantial performance improvements. We will move this discussion to the main paper for clarity.
>
>
> ## SS3: Statistics
>
> We thank the reviewer for this suggestion. We will clarify the motivation for our statistical tests in the paper. Specifically, we use paired tests as each participant interacts with all methods, and apply Holm–Bonferroni correction to control for multiple comparisons across methods per question.
>
> ## SO2: Curriculum and connection to expected improvement.
>
> We thank the reviewer for highlighting this connection. Indeed, the link between learnability and expected policy improvement provides a principled foundation for UPD, and we view this as a key conceptual contribution of the work. Our hope is to build on UPD to develop AHT-UED hybrid methods that generalise to new partners and environments.
>
> ## L1
>
> As noted above, we will include a discussion. Thanks for pointing that out.

---

> > ### Author Rebuttal · Reviewer_ZDAM · 2026-03-31
> >
> > All my flagged weaknesses (WQ2, WQ4, WQ5) have been addressed adequately by the authors, and the limitation L1 is addressable upon camera-ready versioning.
> >
> > I appreciate the paper even more and will raise my scores accordingly.

---

> > > ### Author Response · Authors · 2026-04-01
> > >
> > > Dear ZDAM,
> > >
> > > Thanks again for your kind review and your appreciation of our work. We want to give a minor clarification on runtime numbers. In our rebuttal, we reported FCP runtime (~ 281 min) based on an earlier sequential implementation. However, during the project we developed a parallelised version of FCP using Jax that we used later during our project.  With our current parallelised implementation (e.g., via vmap), FCP requires ~ 80-100 min. This does not affect the qualitative comparison: UPD (~ 51 min) remains faster and avoids population pretraining overhead. We will include these numbers in the final manuscript.

---

### Official Review · Reviewer_8dwv · 2026-03-06

**Soundness:** 3
**Presentation:** 2
**Significance:** 3
**Originality:** 3
**Overall Recommendation:** 5
**Confidence:** 3

**Summary:**

This paper introduces a method, UPD, for joint partner and environment generalization. The key benefit of the approach is that it is population-free, thus providing training partner distributions in a cheap yet effective manner. Experiments showcases the importance of modulating mixture coefficient, a popular AHT mechanism, over the course of agent training. Empirical results show that the method is effective in training robust partner agents.

**Compliance With Llm Reviewing Policy:**

Affirmed.

**Key Questions For Authors:**

- Proposed learnability metric is rather heuristically motivated. This is fine but I was wondering if other metrics were tested out as well, such as TD error, regret, etc.

**Limitations:**

yes

**Strengths And Weaknesses:**

**Strength**
- Sound introduction of UED and its relation to UPD. The method is well-motivated from the frameworks of UED, and appears simple to implement.
- Excellent experiment design and analysis, especially with the analysis on the empirical performance relative to baselines algorithms and qualitative behavior of UPD in emergent convention breaking. Results are consistent across a number of environments and even in joint partner-environment settings.
- Strong results in human experiments as well.

**Weaknesses**
- The algorithmic framework is based only on modulating the mixture coefficient for the specific case of ego-random mixture, even though UPD is posed as a more general framework over partner design space. It would be nice to introduce other variables to modulate, or at least discussions and even preliminary experiments thereof.
- The theoretical impact of section 4.4 is rather limited. This might more appropriately belong in the appendix section

---

> ### Author Rebuttal · Authors · 2026-03-29
>
> Dear 8dwv,
>
> Thank you for your positive and thoughtful review. We are glad that you find the UED perspective well-motivated and the empirical evaluation strong, including the analysis of emergent behaviours and the human studies.
>
> ## Other Variables to Modulate
>
> We agree. In this work, we intentionally instantiate UPD with a simple partner generator (competence mixing + low-level biases) to isolate and validate the core mechanism: learnability-driven partner curricula. A key finding is that even this minimal instantiation already induces non-trivial emergent behaviours (e.g., convention breaking, Fig. 6), supporting our central claim that simple partner generators combined with learnability-driven selection are sufficient to induce structured and adaptive coordination behaviour.
>
> More expressive generators are a natural extension. For example, one could reuse past checkpoints or use learned latent-conditioned generators. Including past checkpoints in the curriculum is especially attractive as it comes with little additional computational overhead and since different checkpoints have been exposed to different conventions (again Fig. 6) and levels of randomness (Fig. 4) they likely introduce diverse behavior effectively. We will expand this discussion in the revision.
>
> ## Section 4.4 might be more appropriate in the appendix.
>
> We agree that Section 4.4 is more conceptual in nature. However, we believe it is important to include it in the main paper as it highlights how the selection mechanism and partner sampling interact and in which way that affects zero-shot coordination. We will streamline the section and move lower-level details to the appendix.
>
> ## Other Learnability Functions
>
> While we did not evaluate TD error or regret directly, we tested several alternative formulations (e.g., Gaussian-weighted scores, coefficient of variation) and found that the method is robust to the choice of learnability metric, as long as it captures variability in outcomes.
>
> From these variance performed best and is most cleanly connected to the results from related work (e.g. expected policy improvement explored in (Foster et al., 2025): They show that expected policy improvement is proportional to the expected gradient magnitude and then connect the expected gradient magnitude to the variance of task outcomes, i.e. learnability as used in e.g. Rutherford et al. (2024).
>
> However, of course not all metrics are suitable: Using mean return as a scoring function for example collapses to self-play, as it favours predictable, self-similar partners (i.e., $\epsilon \rightarrow 0$), eliminating partner diversity.
>
> Overall, these findings suggest that the core mechanism of UPD (learnability-driven partner selection) is robust to design choices and already effective with minimal instantiations, while leaving significant room for future extensions.
>
> We will include a discussion of other UED scores in the revision.

---

> > ### Author Rebuttal · Reviewer_8dwv · 2026-04-03
> >
> > Thank you to the authors for addressing my questions. I will keep my score as it is already fairly positive. Great work!

---

### Official Review · Reviewer_e6kW · 2026-03-15

**Soundness:** 2
**Presentation:** 3
**Significance:** 2
**Originality:** 2
**Overall Recommendation:** 4
**Confidence:** 3

**Summary:**

This paper studies the ad-hoc teamwork problem where an agent has to cooperate with unknown partners during evaluations. The authors propose a population-free partner generation framework (Unsupervised Partner Design) that aims to generate the best mixture of the ego agent's self-policy and a random policy as the partner to maximise the ego agent's learnability. Experiments are conducted in the LBF and Overcooked environments, and results show that UPD consistently improves zero-shot coordination without pre-trained partners or manual tuning when compared against several population-based and population-free baselines.

**Compliance With Llm Reviewing Policy:**

Affirmed.

**Final Justification:**

My concern about the lack of comparison with other partner generation baselines is resolved. The author has pointed out that ROTATE is tuned with task-specific settings, and they have preliminarily shown that a tuned UPD could achieve a comparable or better performance. Together with the computational advantage of UDP, I agree that the proposed UDP is effective. I will raise my score to 4.

**Key Questions For Authors:**

Can you give some insight into why population-based methods are generally better in FC, and why E3T and UPD are much better in CC?

**Limitations:**

yes

**Strengths And Weaknesses:**

Strengths
The analysis is comprehensive, with the chosen epsilons and learnability plots in different game settings. The introduction of the biased random policy is effective. The paper is easy to follow.

Weaknesses
1.	The performance evaluation did not include other partner generation approaches, such as (Li et al., 2023b) and (Wang et al., 2025). Although they maintain a partner population during training, these agents are just a pool of partners generated previously.
2.	The idea of partner generation lacks novelty; the improvement mainly builds on adopting an adaptive epsilon on E3T.

---

> ### Author Rebuttal · Authors · 2026-03-29
>
> Dear e6kW,
>
> We thank you for your constructive review! We are happy to read that you find our analysis to be comprehensive, that our biased random policy introduction is effective and that you find our work to be well written.
>
> ## Other baselines
>
> The core question in this paper is whether explicit partner populations can be removed entirely to enable scalable generalisation across partners and environments. This leads to a different design objective: rather than constructing diverse but task-specific partner populations, we aim to generate appropriate partners online for newly encountered settings efficiently. Our solution is UPD.
>
> We therefore focus on representative baselines across this design space for both fixed-task and procedurally generated settings: population-based AHT, population-free AHT, and joint partner-environment training.
>
> Li et al. (2023b) and Wang et al. (2025) follow population-based approaches where partner sets are expanded over training, optimised for strong performance in a single environment. This distinction is particularly important in procedurally generated settings, where such methods may require increasingly large populations, while UPD naturally scales without maintaining an explicit partner set.
>
> We nevertheless agree that comparing to ROTATE (Wang et al., 2025) is valuable, as it is a strong recent method for the fixed task setting. ROTATE is evaluated with extensive layout-specific hyperparameter sweeps (cf. Table 5 in Wang et al., 2025), including learning rates, entropy coefficients, and PPO parameters. In contrast, UPD uses a shared learning configuration across layouts.
>
> In a preliminary analysis, we report results averaged over 3 seeds per layout. Using this default configuration, UPD achieves higher average performance than ROTATE (94.4 vs. 91.7), while the per-layout comparison is mixed: UPD performs better on AA and CC, whereas ROTATE performs better on CRoom, CR and FC.
>
> Additionally, to assess robustness and make the comparison fairer to UPD, we evaluated a small set of per-layout tuned UPD configurations, varying only key curriculum parameters (buffer size, refresh frequency, rollout factor, number of partners). Under this restricted tuning, UPD achieves higher average performance (102.3 vs. 91.7) and is competitive or better on 4/5 layouts, while remaining below ROTATE on CRoom. UPD is generally robust to hyperparameters (most configurations within 90% of the best).
>
> This tuning is more limited than ROTATE’s sweeps. Despite this, UPD remains strong while requiring significantly less compute (~4-8x faster, ROTATE takes between 3h 30m to 6h 45m on our system, UPD takes ~51m) and no explicit population.
>
> | Method   | CRoom       | AA          | CR          | CC          | FC          | AVG.        | % Gain rel. to E3T |
> |---|---|---|---|---|---|---|----|
> | ROTATE    | 119.8 | 147.7 | 80.5 | 58.0 | 52.1 | 91.6 | 17.6%   |
> | UPD (paper)  | 108.1       | 181.4       | 69.2        | 64.5        | 48.7        | 94.4        | 22.9%  |
> | UPD (tuned) | 112.1     | 201.7     | 79.6      | 66.7      | 51.3      | 102.3    | 28.3%  |
>
> The approaches are not mutually exclusive: UPD could be combined with population-based methods via learnability-based selection and partner editing. We will include a discussion in the revision.
> We will include a discussion in sec. 5, 7 and the appendix.
>
> ## Novelty
>
> We respectfully disagree. Our contribution extends well beyond adapting $epsilon$. We introduce a new perspective on AHT by framing partner generation as an unsupervised curriculum design problem, analogous to UED.
>
> Concretely:
> (i) we introduce a population-free framework for AHT based on learnability-driven partner curricula,
> (ii) we show that this induces an adaptive curriculum over partners at the learning frontier (Fig. 4-6), and
> (iii) we extend this to the joint partner-environment setting, enabling generalisation to unseen layouts.
>
> A key empirical finding is that this instantiation (combining a lightweight partner generator with a learnability-based curriculum) already induces non-trivial emergent behaviours (e.g., convention breaking, Fig. 6) as a result of the interaction between the partner generator and the selection mechanism.
>
> Finally, our design objective (a mechanism for generalisation and adaptation to millions of tasks) is fundamentally different from prior approaches. The adaptive $\epsilon$ mechanism is one instantiation within this broader framework. Please also see our previous answer.
>
> ## Results on CC and FC
> Population-based methods are not generally weak on CC (Fig. 11–12 show convergence). However, CC is particularly challenging, and we observe that FCP and MEP rely more strongly on their training partners and may exhibit freeriding. ROTATE does not show this behaviour but still underperforms UPD.
> For FC, the set of viable strategies is narrow and can be well covered by a population. With additional tuning, UPD becomes competitive or better (see above).

---

> > ### Author Rebuttal · Reviewer_e6kW · 2026-04-04
> >
> > Thank you for the authors’ reply.
> >
> > My concern about the lack of comparison with other partner generation baselines is resolved. The author has pointed out that ROTATE is tuned with task-specific settings, and they have preliminarily shown that a tuned UPD could achieve a comparable or better performance. Together with the computational advantage of UDP, I agree that the proposed UDP is effective. I will raise my score to 4.
> >
> > I encourage the authors to include the comparison in the appendix.

---

### Decision · Program_Chairs · 2026-04-30

**Decision:**

Accept (spotlight)

**Comment:**

The paper offers a solid contribution for the ad-hoc teamwork setting.  The authors effectively addressed reviewer questions in the rebuttal stage, which increased reviewer enthusiasm for the paper.